# CURV: Coherent Uncertainty-Aware Reasoning in Vision-Language Models for X-Ray Report Generation

**Ziao Wang**[1,2], **Sixing Yan**[1], **Kejing Yin**[1]*, **Xiaofeng Zhang**[3], **William K. Cheung**[1]

[1]Department of Computer Science, Hong Kong Baptist University
[2]Institute of Systems Medicine and Health Sciences, Hong Kong Baptist University
[3]Department of Computer Science, Harbin Institute of Technology

## Abstract

Vision-language models have been explored for radiology report generation with promising results. Yet, uncertainty elaborated in findings and the reasoning process for reaching clinical impressions are seldom explicitly modeled, reducing the clinical accuracy and trustworthiness of the generated reports. We present CURV, a novel framework that alleviates the limitations through integrated awareness of uncertainty and explicit reasoning capabilities. Our approach consists of three key components: (1) an uncertainty modeling mechanism that teaches the model to recognize and express appropriate levels of diagnostic confidence, (2) a structured reasoning framework that generates intermediate explanatory steps connecting visual findings to clinical impressions, and (3) a reasoning coherence reward that ensures logical consistency among findings, reasoning, and impressions. We implement CURV through a three-stage training pipeline that combines uncertainty-aware fine-tuning, reasoning initialization, and reinforcement learning. In particular, we adopt a comprehensive reward function that addresses multiple aspects of report quality, incorporating medical term matching, uncertainty expression evaluation, and semantic coherence evaluation. Experimental results demonstrate that CURV generates clinically relevant reports with appropriate uncertainty expressions and transparent reasoning traces, significantly outperforming previous methods. CURV[2] represents a substantial advancement toward interpretable and trustworthy AI-generated radiology reports, with broader implications for the deployment of vision-language models in high-stakes clinical environments where uncertainty awareness and reasoning transparency are essential.

## 1 Introduction

Chest X-rays (CXRs) are a cornerstone of diagnostic imaging, yet their interpretation is time-intensive, straining healthcare systems amid radiologist shortages. Automated report generation using vision-language models (VLMs) offers a promising solution to enhance efficiency and reduce workload [31, 20]. Moreover, this technology can help bridge the gap in report quality between large and small hospitals, where physicians in smaller facilities often have less experience compared to those in major centers, enabling more consistent and accurate diagnoses across diverse settings.

Medical report generation, unlike general image captioning tasks, imposes unique challenges that current VLMs are not fully equipped to address, particularly in handling diagnostic uncertainty and providing transparent reasoning [35]. Radiologists routinely employ linguistic markers such as "likely"

---

*Correspondence to: Kejing Yin <cskjyin@comp.hkbu.edu.hk>
[2]Code and Dataset: `https://github.com/wwwadx/CURV`

39th Conference on Neural Information Processing Systems (NeurIPS 2025).

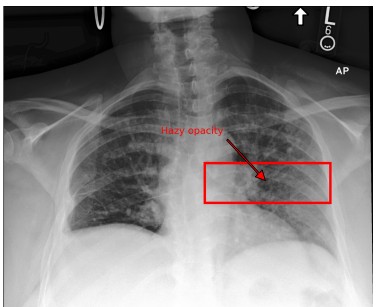

**Structural Uncertainty (Findings):** *"Pulmonary nodules in the left upper lobe are also **not completely characterized** on this study. However, in addition, there is a more hazy widespread opacity projecting over the left mid upper lung which **could be compatible with** a coinciding pneumonia."*

**Semantic Uncertainty (Impression):** *"Increasing left lung opacification which **may reflect** pneumonia superimposed on metastatic disease, although other etiologies such as lymphangitic pattern of metastatic spread **could be considered**. CT **may be helpful** to evaluate further if needed clinically."*

(a) Chest X-ray image      (b) Corresponding uncertain expressions from the radiology report.

Figure 1: Illustration of Structural and Semantic Uncertainty. (a) A hazy opacity in the left mid-upper lung (arrow) generates structural uncertainty regarding its nature. (b) This feeds into the semantic uncertainty in the impression, where multiple etiologies are considered and further investigation (CT) is suggested.

or "possible" to convey varying degrees of diagnostic confidence, ensuring clear communication with referring physicians [32]. As diagnostic uncertainty is multifaceted, as illustrated in Figure 1, it is crucial to distinguish between at least two key types: (1) **Structural Uncertainty** regarding specific visual findings (e.g., a sentence in the "Findings" section stating "hazy opacity... *could be compatible with* pneumonia"); and (2) **Semantic Uncertainty** stated in the overall "Impression" (e.g., "opacification *may reflect* pneumonia... although other etiologies *could be* considered"). Accurately modeling these uncertainties, and the reasoning that connects them, is crucial for generating trustworthy reports to support effective clinical decision-making. Crucially, our work frames this challenge not as quantifying a model's internal statistical confidence, but as modeling the linguistic expression of diagnostic uncertainty—the specific language radiologists use to convey confidence levels. Moreover, robust clinical decision-making relies heavily on explicit reasoning that logically connects such visual findings, with their structural uncertainties, to the more diagnosis-oriented medical impressions and their associated semantic uncertainties.

Existing VLMs for CXR report generation often prioritize factual accuracy over modeling diagnostic uncertainty or providing explicit reasoning pathways, resulting in reports that lack clinical nuance and transparency [41, 29, 8, 38, 23]. This deficiency poses a significant barrier to their clinical adoption, as physicians require both accurate uncertainty expression and transparent reasoning to trust and effectively utilize AI-generated reports. Consequently, there is a pressing need for developing approaches that address these critical gaps by integrating robust uncertainty awareness and transparent reasoning mechanisms into the report generation process.

To tackle these challenges, we introduce CURV, a novel framework for uncertainty-aware vision-language models with explicit reasoning capabilities for CXR report generation. CURV advances the field through three key innovations: (1) a systematic uncertainty modeling approach that enables the model to recognize anatomical structures and express appropriate diagnostic confidence using a data-driven fine-tuning strategy and a specialized uncertainty reward; (2) a structured reasoning framework that generates intermediate explanatory steps linking visual findings to clinical impressions, thereby enhancing transparency. To enable the development and supervised initialization of this reasoning capability, we created **TRACE-CXR** (Transparent Reasoning and Articulation for Clinical Explanations - CXR), a novel dataset of 2,000 chest X-ray reports, each augmented with an explicit, LLM-generated "thinking" section that models the reasoning pathway from findings to impression; and (3) a multi-dimensional reward design in reinforcement learning that ensures logical consistency across findings, reasoning, and impressions, addressing the need for coherent and trustworthy reports. Our approach leverages a vision language model, augmented by a meticulously designed training pipeline that embeds uncertainty awareness and reasoning capabilities directly into the model. The uncertainty modeling component is developed through targeted fine-tuning with uncertainty-annotated data, enabling the model to predict confidence levels for identified pathologies and map them to appropriate linguistic expressions in the generated text. For the reasoning component, we employ a structured generation process that articulates visual findings, produces intermediate reasoning steps, and delivers clinical impressions—all while maintaining logical coherence across these elements through a novel optimization strategy.

The contributions of this study are as follows:

- We propose a framework for uncertainty-aware medical report generation, integrating a specialized fine-tuning strategy with curated uncertainty-annotated data and uncertainty-calibrated reward mechanism to enhance the clinical relevance of AI-generated CXR reports.
- We introduce a structured reasoning framework that leverages our **TRACE-CXR** dataset to initialize explicit "thinking" pathways within a tripartite report structure (findings, thinking, impression), supported by a multi-dimensional reward design in reinforcement learning, to ensure transparent explanations and logical consistency between radiological observations and clinical impressions.
- Through extensive experimentation, we demonstrate that CURV generates clinically relevant CXR reports with appropriate uncertainty expressions and transparent reasoning traces, outperforming existing methods in both qualitative and quantitative metrics, thus advancing trustworthy AI in high-stakes clinical environments.

## 2 Related Work

### 2.1 Uncertainty in Medical Report Generation

Expressing diagnostic uncertainty is critical in radiology for clear communication and effective clinical decision-making. Equipping vision-language models (VLMs) with uncertainty awareness is thus essential for clinical adoption. Existing studies have approached this from various perspectives. Wang et al. [32] used Monte Carlo dropout to estimate visual and textual uncertainty, integrating it into a weighted loss function for reliable outputs. Similarly, Yan et al. [35] introduced the Diagnostic Uncertainty Encoding framework to encode clinically inspired uncertainty concepts, enhancing report accuracy. Najdenkoska et al. [22] proposed a probabilistic latent variable model with variational topic inference to generate diverse CXR reports reflecting multiple interpretations. Additionally, Yang et al. [37] demonstrated that training Bayesian neural networks with uncertain labels increases predictive variance for ambiguous cases, while large-scale VLMs like Med-Gemini [25] employ uncertainty-guided strategies for clinical reasoning tasks. However, many approaches lack explicit differentiation between uncertainty in specific findings (Structural Uncertainty) and overall diagnostic synthesis (Semantic Uncertainty), as noted by [41, 29]. CURV addresses this gap by systematically modeling and expressing both types of uncertainty through a specialized reward mechanism and fine-tuning with uncertainty-annotated data, aiming for clinically nuanced reports.

### 2.2 Reasoning by Reinforcement Learning

Reinforcement learning (RL) has shown promise in enhancing reasoning capabilities in language and vision-language models [40, 18, 34]. DeepSeek-R1 [7] leverages Group Relative Policy Optimization (GRPO) [28] to develop reasoning skills via rule-based rewards. Extending RL to VLMs, Huang et al. [12] introduced Vision-R1, achieving strong performance through automated dataset construction and Progressive Thinking Suppression Training with GRPO [26, 39]. In the medical domain, RL is increasingly applied to radiology report generation for structured reasoning and trustworthiness [11, 42]. Jing et al. [15] proposed BoxMed-RL, focusing on explainable reports through CoT reasoning and spatially verifiable RL, emphasizing spatial grounding via IoU-based rewards. Similarly, Shao et al. [27] used RL to improve alignment between radiology images and reports, targeting linguistic quality and anomaly detection. While these works highlight RL's potential for logical consistency and transparency, CURV uniquely integrates uncertainty modeling with structured reasoning, using a multi-dimensional reward design to ensure coherent and trustworthy AI-generated reports.

## 3 The Proposed Method

The CURV framework, detailed in this section, is designed to produce radiology reports that embody both diagnostic accuracy and a sophisticated understanding of clinical uncertainty. Our method explicitly models the *Structural Uncertainty* (tied to individual findings) and *Semantic Uncertainty* (concerning the overall diagnostic impression), along with the coherent reasoning that bridges them.

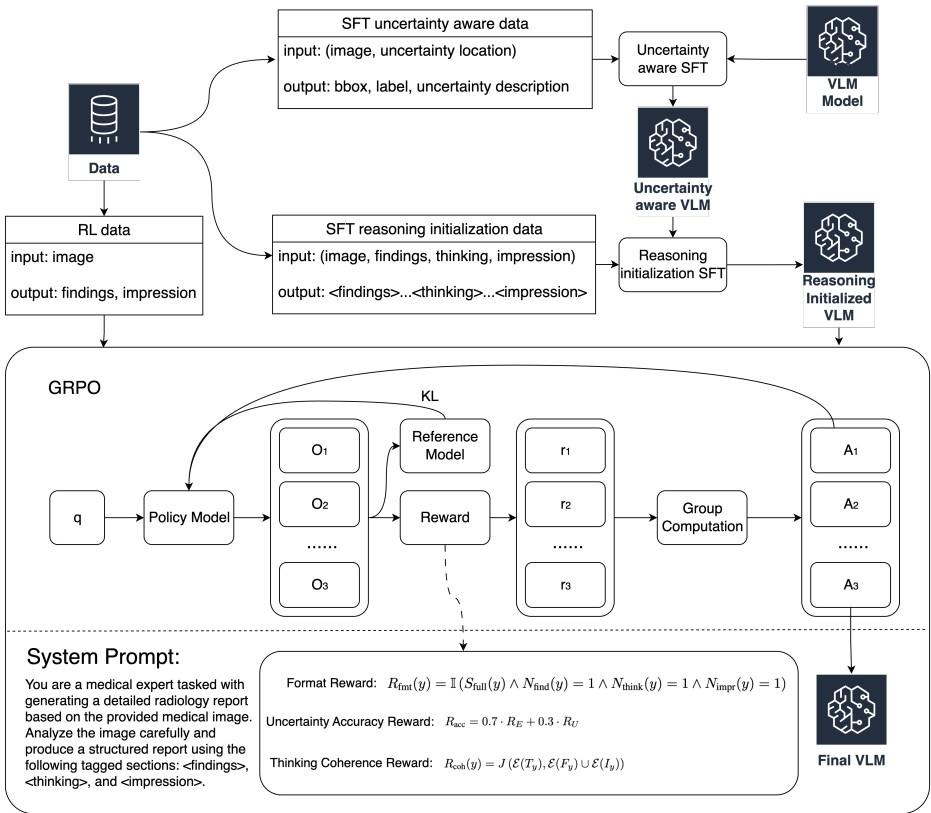

Figure 2: The CURV framework architecture. The figure illustrates the three-stage training pipeline, beginning with SFT for uncertainty awareness, followed by reasoning initialization, and culminating in a Reinforcement Learning phase using GRPO to refine the final Vision-Language Model.

## 3.1 Problem Formulation

The generation of medical reports from chest X-ray (CXR) images using vision-language models (VLMs) is a complex task requiring diagnostic accuracy, uncertainty expression, and transparent reasoning. Formally, given a CXR image $I$ and a prompt $p$ (e.g., requesting a structured report), the goal is to train a VLM $\pi_\theta$ to generate a structured output $y$ with three components: visual findings, logical reasoning, and clinical impressions with uncertainty expressions. The objective is to optimize report quality across accuracy, uncertainty awareness, and coherence, formulated as:

$$\theta^* = \arg\max_\theta \mathbb{E}_{(I,p)\sim\mathcal{D}, y\sim\pi_\theta}[r(y, I, p)], \tag{1}$$

where $\mathcal{D}$ is the dataset of image-prompt pairs, $y$ is the generated report, and $r(y, I, p)$ is a multi-dimensional reward function assessing format adherence, medical accuracy, uncertainty expression, and reasoning coherence. This framework, CURV, aims to produce clinically relevant, transparent, and trustworthy AI-generated reports by addressing these critical dimensions in a unified manner.

## 3.2 Empowering Uncertainty Awareness in Vision-Language Models

Current vision-language models (VLMs) often fail to capture the inherent uncertainty in medical report generation, a critical limitation for clinical decision-making in radiology. The CURV framework addresses this gap by instilling awareness of *Structural Uncertainty*, which pertains to specific findings, as introduced in Section 1. This initial stage focuses on enabling the model to recognize and articulate appropriate confidence levels for anatomical abnormalities, thereby enhancing the reliability of the "Findings" section in generated reports. The supervised fine-tuning (SFT) process leverages a curated uncertainty-annotated dataset $\mathcal{D}$, where each instance includes a CXR image $I$, a prompt $p$ (e.g., instructing detection of anatomical objects with uncertainties), and ground-truth structural

uncertainty specifications $Y_{gt} = \langle f_1, f_2, \ldots, f_{N_I} \rangle$. Each finding $f_k = (b_k, l_k, t_k)$ encapsulates a bounding box, anatomical label, and textual uncertainty description. The model $\pi_\theta$ is trained to generate a serialized token sequence $\text{seq}(Y_{gt})$ using the loss function:

$$\mathcal{L}_{\text{uncertainty}} = - \sum_{(I,p,Y_{gt}) \in \mathcal{D}} \log \pi_\theta(\text{seq}(Y_{gt})|I,p), \tag{2}$$

where $\log \pi_\theta(\text{seq}(Y_{gt})|I,p)$ is the log-probability of producing the target sequence encoding all structural findings. This loss guides the model to detect uncertain regions (via $b_k$), identify them (via $l_k$), and articulate specific *Structural Uncertainty* (via $t_k$) in a structured, sequentially generated format.

### 3.3 Reasoning Initialization for Transparent Reporting

Building on the uncertainty awareness developed earlier, the second stage of CURV focuses on initializing basic reasoning capabilities in the model. The objective is to generate transparent reports by articulating logical connections between radiological findings and clinical impressions through an intermediate reasoning path, enhancing interpretability for clinical validation.

To achieve this, a report generation task is structured with three components: Findings ($F$), Reasoning ($R$), and Impression ($C$), with $R$ elucidating the transition $F \xrightarrow{R} C$ to mimic a radiologist's inferential steps. Inspired by frameworks like DeepSeek-R1 [7], which use structured data to improve reasoning patterns, this stage employs supervised fine-tuning with a dataset we developed for this purpose, $\mathcal{D}_{\text{reason}}$. Each instance includes a CXR image $I$, a prompt $p$ (e.g., "Generate a detailed radiology report"), and a ground-truth structured report $Y_{\text{structured}} = \langle \text{text}_F, \text{text}_R, \text{text}_C \rangle$. Here, $\text{text}_R$ provides a logical narrative linking observations in $\text{text}_F$ to conclusions in $\text{text}_C$, detailing abnormalities, suspected conditions, differential diagnoses, and contextual information. The VLM $\pi_\theta$ is fine-tuned on $\mathcal{D}_{\text{reason}}$ to maximize the log-likelihood of the serialized sequence $\text{seq}(Y_{\text{structured}})$, with the loss defined as:

$$\mathcal{L}_{\text{reasoning}} = - \sum_{(I,p,Y_{\text{structured}}) \in \mathcal{D}_{\text{reason}}} \log \pi_\theta(\text{seq}(Y_{\text{structured}})|I,p). \tag{3}$$

By training the model with $\mathcal{L}_{\text{reasoning}}$, we instill a foundational capability to generate reports that are not only descriptive but also explanatory, paving the way for more sophisticated coherence and alignment in the subsequent reinforcement learning stage.

### 3.4 Enhancing Clinical Reasoning with Reinforcement Learning

Building upon the foundational capabilities empowered by the previous stages for uncertainty awareness and structured reasoning, the final CURV stage employs Reinforcement Learning (RL) to refine the model's clinical reasoning process. Our SFT-then-RL methodology is designed to move beyond simple imitation [6]. The preceding SFT stage uses the TRACE-CXR dataset to teach the model the basic tripartite report structure, providing a foundational policy for exploration. In this RL phase, however, the model is trained on the MIMIC-CXR dataset and is provided with only the ground-truth findings and impression sections. This design avoids the "imitation trap" where models simply reproduce potentially flawed or suboptimal reasoning paths from the SFT data. Instead, it forces the model to discover a functionally coherent reasoning process on its own, guided only by the reward signals that measure the logical connection between the human-authored findings and impressions. This ensures the model learns to generate genuinely coherent reasoning rather than engaging in "pseudo reasoning" by merely mimicking a template [4].

To this end, we use Group Relative Policy Optimization (GRPO) [28], guided by a novel, multi-component reward function tailored for clinical report generation. A key advantage of this RL phase, particularly through the coherence reward ($R_{\text{coh}}$), is its ability to guide the model in generating the intermediate "thinking" process that logically connects findings to impressions, without requiring extensive supervised data for this specific reasoning component.

Given an input $(I,p)$ and ground truth $y_{\text{gt}}$, GRPO samples $G$ reports $\{y_g\}$ from policy $\pi_{\theta_{\text{old}}}$ and updates $\pi_\theta$ by maximizing:

$$J(\theta) = \mathbb{E}\left[\frac{1}{G}\sum_{g=1}^{G}\left(\min(\rho_g A_g, \text{clip}(\rho_g, 1-\epsilon, 1+\epsilon)A_g)\right) - \beta D_{KL}(\pi_\theta||\pi_{\text{ref}})\right], \tag{4}$$

where $J(\theta)$ represents the objective function, which aims to maximize the expected reward for the policy $\pi_\theta$ while maintaining stability in training. The function incorporates a clipped advantage term to limit large policy updates and a KL-divergence penalty term ($D_{KL}$) with coefficient $\beta$ to prevent excessive deviation from a reference policy $\pi_{\mathrm{ref}}$. The expectation $\mathbb{E}$ is taken over sampled reports, with $G$ denoting the number of sampled reports $\{y_g\}$ from the old policy $\pi_{\theta_{\mathrm{old}}}$. And $\rho_g$ is the importance sampling ratio and $A_g = R_{\mathrm{total}}(y_g, y_{\mathrm{gt}}) - \mathrm{mean}(\{R_{\mathrm{total}}(y_k, y_{\mathrm{gt}})\})$ is the advantage. The total reward $R_{\mathrm{total}}(y, y_{\mathrm{gt}})$ is a weighted sum of three components:

$$R_{\mathrm{total}}(y, y_{\mathrm{gt}}) = w_{\mathrm{fmt}} R_{\mathrm{fmt}}(y) + w_{\mathrm{acc}} R_{\mathrm{acc}}(y, y_{\mathrm{gt}}) + w_{\mathrm{coh}} R_{\mathrm{coh}}(y). \tag{5}$$

The total reward $R_{\mathrm{total}}(y, y_{\mathrm{gt}})$ evaluates the quality of a generated report $y$ compared to the ground truth $y_{\mathrm{gt}}$ by combining three distinct reward components with corresponding weights $w_{\mathrm{fmt}}$, $w_{\mathrm{acc}}$, and $w_{\mathrm{coh}}$. These components assess format adherence ($R_{\mathrm{fmt}}$), medical accuracy and uncertainty alignment ($R_{\mathrm{acc}}$), and reasoning coherence ($R_{\mathrm{coh}}$), respectively, ensuring a comprehensive evaluation of clinical report quality.

**Format Adherence Reward ($R_{\mathbf{fmt}}$):**    A binary reward $\mathbb{I}(\cdot)$ verifying strict adherence to the tripartite structure ($\langle\mathrm{findings}\rangle, \langle\mathrm{thinking}\rangle, \langle\mathrm{impression}\rangle$), ensuring each tag appears exactly once and in order, with no extraneous content.

$$R_{\mathrm{fmt}}(y) = \mathbb{I}\left(S_{\mathrm{full}}(y) \wedge N_{\mathrm{find}}(y) = 1 \wedge N_{\mathrm{think}}(y) = 1 \wedge N_{\mathrm{impr}}(y) = 1\right), \tag{6}$$

where $R_{\mathrm{fmt}}(y)$ is a binary indicator function $\mathbb{I}(\cdot)$ that returns 1 only if the generated report $y$ fully adheres to the required structure. Specifically, $S_{\mathrm{full}}(y)$ checks if all required sections are present, while $N_{\mathrm{find}}(y) = 1$, $N_{\mathrm{think}}(y) = 1$, and $N_{\mathrm{impr}}(y) = 1$ ensure that each section (findings, thinking, impression) appears exactly once in the correct order.

**Findings/Impression Uncertainty Accuracy Reward ($R_{\mathbf{acc}}$):**    Evaluates medical accuracy and uncertainty in findings ($F_y$) and impression ($I_y$) sections against $y_{\mathrm{gt}}$, weighted as $R_{\mathrm{acc}} = 0.7 \cdot R_E + 0.3 \cdot R_U$. Returns 0 if sections are missing.

1. **Entity Matching ($R_E$):** Average F1-score of RadGraph-extracted entities [14] between $F_y, I_y$ and $F_{y_{\mathrm{gt}}}, I_{y_{\mathrm{gt}}}$.

$$R_E(y, y_{\mathrm{gt}}) = \frac{1}{2}\left(\mathrm{F1}(\mathcal{E}(F_y), \mathcal{E}(F_{y_{\mathrm{gt}}})) + \mathrm{F1}(\mathcal{E}(I_y), \mathcal{E}(I_{y_{\mathrm{gt}}}))\right), \tag{7}$$

   where $R_E(y, y_{\mathrm{gt}})$ computes the average F1-score for entity matching between the generated report sections ($F_y$ for findings and $I_y$ for impression) and the ground truth sections ($F_{y_{\mathrm{gt}}}$ and $I_{y_{\mathrm{gt}}}$). The function $\mathcal{E}(\cdot)$ extracts entities using RadGraph, and $\mathrm{F1}(\cdot, \cdot)$ measures the overlap between entity sets, equally weighting the performance on findings and impression sections.

2. **Uncertainty Alignment ($R_U$):** Average alignment of (entity, term, score) triples $\mathcal{P}(S_{y'})$ between sections.

$$R_U(y, y_{\mathrm{gt}}) = \frac{1}{2}\left(\mathrm{MatchPairs}(\mathcal{P}(F_y), \mathcal{P}(F_{y_{\mathrm{gt}}})) + \mathrm{MatchPairs}(\mathcal{P}(I_y), \mathcal{P}(I_{y_{\mathrm{gt}}}))\right), \tag{8}$$

   where $R_U(y, y_{\mathrm{gt}})$ measures the alignment of uncertainty expressions between the generated report sections ($F_y$ and $I_y$) and the ground truth ($F_{y_{\mathrm{gt}}}$ and $I_{y_{\mathrm{gt}}}$). The function $\mathcal{P}(\cdot)$ extracts (entity, term, score) triples, and $\mathrm{MatchPairs}(\cdot, \cdot)$ evaluates their similarity, averaging the results for findings and impression sections. MatchPairs combines semantic term similarity (0.4 weight) and score difference (0.6 weight) for common entities, then combines this average similarity (0.7 weight) with entity coverage (0.3 weight).

**Thinking Coherence Reward ($R_{\mathbf{coh}}$):**    Measures Jaccard similarity $J(\cdot, \cdot)$ between RadGraph-extracted entities in the thinking section $\mathcal{E}(T_y)$ and the union of entities in findings $\mathcal{E}(F_y)$ and impression $\mathcal{E}(I_y)$. Returns 0 if sections are missing or entity sets are empty.

$$R_{\mathrm{coh}}(y) = J\left(\mathcal{E}(T_y), \mathcal{E}(F_y) \cup \mathcal{E}(I_y)\right), \tag{9}$$

where $R_{\mathrm{coh}}(y)$ quantifies the coherence of the thinking section ($T_y$) in the generated report $y$ by computing the Jaccard similarity $J(\cdot, \cdot)$ between entities extracted from $T_y$ (via $\mathcal{E}(T_y)$) and the union of entities from the findings ($\mathcal{E}(F_y)$) and impression ($\mathcal{E}(I_y)$) sections. This ensures that the reasoning process logically connects observations to conclusions. This multifaceted RL optimization aims for reports that are structurally sound, factually accurate with appropriate uncertainty, and demonstrate transparent clinical reasoning.

# 4 Experiments and Analysis

## 4.1 Experimental Setup

We conducted experiments on 4xA100 GPUs using Qwen-2.5-VL-3B as the backbone model, balancing efficiency and performance. Training spanned multiple stages over approximately 100 hours with a batch size of 16 and a learning rate of $1 \times 10^{-6}$. CURV was benchmarked against established vision-language models like LLaVA-1.5-7B and MAIRA-2 under consistent conditions. Full details on configurations and baseline setups are provided in Appendix A.

## 4.2 Datasets

Our experiments leverage the MIMIC-CXR dataset. As a key contribution of this work, we curated specialized data subsets: (1) an uncertainty-annotated dataset with 112,111 samples, and (2) our novel **TRACE-CXR** dataset (**T**ransparent **R**easoning and **A**rticulation for **C**linical **E**xplanations in CXR), featuring 2,000 reports with explicit reasoning pathways. Both were developed to support uncertainty modeling and structured reasoning, respectively, significantly improving data utility. The clinical validity of our TRACE-CXR dataset was subsequently confirmed through a formal evaluation with a board-certified radiologist, which revealed a strong concordance with expert judgment (see Appendix B). Detailed curation processes and statistics for both datasets are also described in Appendix B.

## 4.3 Evaluation Metric

We assess CURV using standard NLP metrics (e.g., BLEU, ROUGE-L, METEOR) for textual quality and clinical accuracy metrics (e.g., CheXbert, RadGraph F1-scores) for medical relevance. Additionally, LLM-based evaluation protocols are employed to evaluate the unique "Thinking" section and uncertainty expressions (structural and semantic). Complete metric definitions and evaluation protocols are available in Appendix C.

Table 1: Generation metrics for radiology report generation across different models

| Model | B-1 | B-2 | B-3 | B-4 | METEOR | R-L | gritlm |
|---|---|---|---|---|---|---|---|
| LLaVA-1.5-7B [21] | 19.09 | 7.46 | 2.81 | 1.25 | 19.16 | 18.36 | 44.25 |
| LLaVA-1.5-7B-SFT-CXR | 22.58 | 15.06 | 9.43 | 6.13 | 25.71 | 28.09 | 50.28 |
| HuatuoGPT-Vision-7B [5] | 19.33 | 9.42 | 4.64 | 1.93 | 26.01 | 20.78 | 47.32 |
| MAIRA-2 [3] | 24.94 | 14.12 | 9.01 | 6.14 | 26.78 | 28.65 | 47.48 |
| Qwen2.5-VL-3B [1] | 13.09 | 5.42 | 2.08 | 0.89 | 20.81 | 15.23 | 44.57 |
| Gemini 2.5 pro | 12.54 | 5.20 | 2.25 | 1.05 | 21.19 | 15.01 | 40.41 |
| CURV_stage1 | 14.96 | 7.08 | 3.33 | 1.61 | 23.47 | 19.07 | 45.15 |
| CURV_stage2 | 10.72 | 5.22 | 2.43 | 1.10 | 18.57 | 14.59 | 42.76 |
| **CURV** | **25.38** | **15.58** | **9.85** | **6.18** | **30.43** | **31.19** | **50.48** |

Table 2: Clinical accuracy metrics for radiology report generation across different models

| Model | CheXbert | | | RadGraph | |
|---|---|---|---|---|---|
| | Acc. | Macro F1 | Micro F1 | Ent. F1 | F1 |
| LLaVA-1.5-7B [21] | 63.25 | 4.94 | 38.54 | 7.61 | 4.95 |
| LLaVA-1.5-7B-SFT-CXR | 72.72 | 5.00 | 51.51 | 17.57 | 13.06 |
| HuatuoGPT-Vision-7B [5] | 71.15 | 5.34 | 48.62 | 15.92 | 9.06 |
| MAIRA-2 [3] | 67.39 | **6.34** | 46.53 | 25.01 | 17.05 |
| Qwen2.5-VL-3B [1] | 67.78 | 4.75 | 37.66 | 9.46 | 4.66 |
| Gemini 2.5 pro | 74.35 | 5.35 | 48.45 | 13.09 | 7.71 |
| CURV_stage1 | 57.59 | 4.51 | 30.75 | 17.00 | 9.95 |
| CURV_stage2 | 56.57 | 3.87 | 26.83 | 11.11 | 6.03 |
| **CURV** | **76.93** | 5.22 | **57.12** | **25.95** | **19.54** |

## 4.4 Main Results

**Overall Metric** The experimental results in Tables 1 and 2 highlight the effectiveness of the CURV framework in radiology report generation, with notable insights from both metric interpretations and model scale perspectives. From the metrics' standpoint, CURV achieves superior generation quality with top scores in BLEU (e.g., BLEU-3: 9.85), METEOR (30.43), and ROUGE_L (31.19) compared to SOTA method like MAIRA-2 (BLEU-3: 9.01), indicating enhanced fluency and textual relevance. Likewise, CURV's ability to produce medically relevant content is underscored by its strong clinical accuracy. This advantage is particularly pronounced when compared to frontier generalist models; while such models are powerful, CURV's specialized approach significantly outperforms Gemini 2.5 Pro across all clinical metrics, achieving a RadGraph F1 score of 19.54 versus 7.71. This demonstrates the critical value of a targeted framework for this complex medical task. Finally, from a model scale perspective, CURV, with a compact 3B parameter size, outperforms larger 7B models like HuatuoGPT-Vision-7B and MAIRA-2, demonstrating that our proposed method enables efficient performance without requiring extensive computational resources.

Table 3: Generation metrics for radiology report generation on IU X-ray dataset

| Model | B-1 | B-2 | B-3 | B-4 | METEOR | R-L | gritlm |
|---|---|---|---|---|---|---|---|
| LLaVA-1.5-7B | 16.52 | 6.60 | 3.00 | 1.40 | 19.65 | 17.48 | 45.78 |
| LLaVA-1.5-7B-SFT-CXR | 21.42 | 12.95 | 8.03 | 5.20 | 23.24 | 26.40 | 46.21 |
| HuatuoGPT-Vision-7B | 19.33 | 10.70 | 6.28 | 2.81 | 31.02 | 23.42 | 50.22 |
| MAIRA-2 | 26.37 | 15.60 | 9.64 | 6.03 | 25.52 | 31.18 | 54.18 |
| Qwen2.5-VL-3B | 11.38 | 5.05 | 2.28 | 1.05 | 21.65 | 15.01 | 45.95 |
| CURV_stage1 | 12.51 | 6.24 | 3.18 | 1.57 | 23.64 | 18.18 | 46.76 |
| CURV_stage2 | 10.18 | 5.32 | 2.75 | 1.27 | 18.64 | 13.93 | 44.46 |
| **CURV** | **29.23** | **18.76** | **12.08** | **6.86** | **38.30** | **39.08** | **54.89** |

Table 4: Clinical accuracy metrics for radiology report generation on IU X-ray dataset

| Model | CheXbert | | | RadGraph | |
|---|---|---|---|---|---|
| | Acc. | Macro F1 | Micro F1 | Ent. F1 | F1 |
| LLaVA-1.5-7B | 72.03 | 4.66 | 46.81 | 13.37 | 8.76 |
| LLaVA-1.5-7B-SFT-CXR | 76.34 | 5.84 | 53.33 | 16.19 | 10.31 |
| HuatuoGPT-Vision-7B | 89.89 | 5.72 | 67.07 | 22.98 | 13.96 |
| MAIRA-2 | 88.74 | **6.22** | 70.75 | 34.53 | 24.01 |
| Qwen2.5-VL-3B | 80.64 | 4.92 | 49.47 | 12.70 | 6.26 |
| CURV_stage1 | 68.89 | 4.57 | 40.30 | 20.59 | 12.13 |
| CURV_stage2 | 67.59 | 3.76 | 33.67 | 12.83 | 7.28 |
| **CURV** | **91.56** | 5.86 | **74.36** | **36.99** | **25.65** |

**Out-of-Distribution Evaluation** To assess the framework's generalization capabilities, we conducted a rigorous out-of-distribution (OOD) evaluation using the IU X-ray [9] dataset, which was not used during training. As shown in the results (Tables 3 and Tables 4), CURV maintains its strong performance, outperforming all baseline methods on this new dataset. This successful performance demonstrates that the model's learned capabilities for reasoning and expressing uncertainty are robust and can generalize well beyond the MIMIC-CXR dataset it was trained on. These findings provide strong evidence that CURV is not overfit to its training data and can be effectively applied to different clinical data sources.

**LLM-based Evaluation** The LLM-based evaluation, as shown in Figure 3, demonstrates substantial qualitative improvements in CURV-generated reports throughout the reinforcement learning phase. This evaluation is based on criteria defined in Appendix C. The "Thinking" section shows consistent and significant enhancements across all assessed criteria—*Logical Coherence*, *Depth of Analysis*, *Relevance*, *Evidence-Based Nature*, and *Consistency*—with the corresponding scores progressively rising (e.g., *Logical Coherence* from 2.23 to 4.55 and *Consistency* from 2.88 to 4.76 by step 3,200).

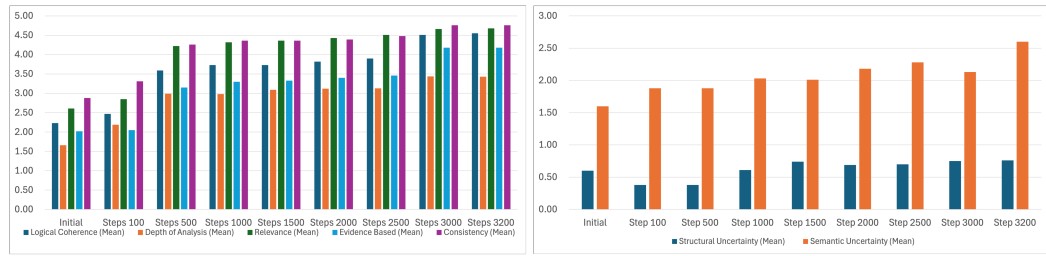

(a) LLM-based thinking evaluation        (b) LLM-based uncertainty evaluation

Figure 3: LLM-based evaluation of report quality during reinforcement learning. (a) shows the progressive improvement of the 'Thinking' section across five qualitative criteria. (b) tracks the increasing scores for both structural and semantic uncertainty expression, demonstrating the model's refinement over training steps.

This indicates the model's increasing ability to articulate a transparent and sound reasoning process. Regarding uncertainty expression, Semantic Uncertainty scores steadily improved from 1.60 to 2.60, indicating better conveyance of overall diagnostic confidence. Structural Uncertainty scores also increased from an initial value of 0.60 to 0.76 (following an early dip), signifying progress in articulating confidence for specific findings, albeit with more complex learning dynamics observed. Collectively, these trends underscore the efficacy of the CURV framework, especially its RL stage, in fostering clinically nuanced reports that are more interpretable, trustworthy, and adept at expressing diagnostic uncertainty, thereby enhancing clinical utility.

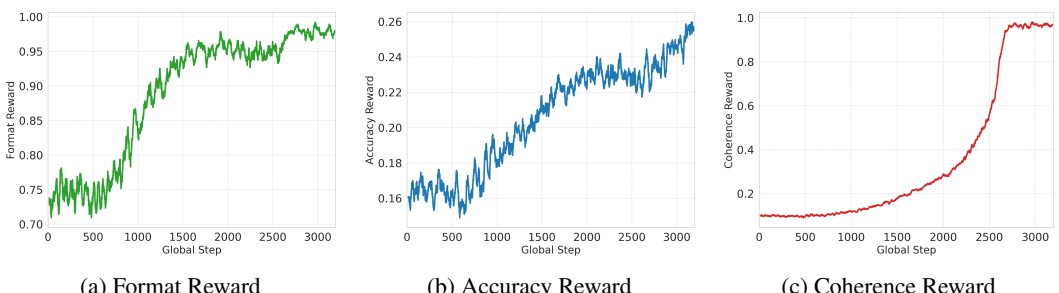

(a) Format Reward        (b) Accuracy Reward        (c) Coherence Reward

Figure 4: Reward trends during the reinforcement learning phase. The plots show the rapid convergence of the (a) Format Reward, the steady increase of the (b) Accuracy Reward, and the significant late-stage ascent of the (c) Coherence Reward, validating the effectiveness of the multi-component reward function.

**Reward Changing**     The evolution of reward components during reinforcement learning, as depicted in Figure 4, underscores the efficacy of the CURV training strategy. The Format Adherence Reward ($R_{fmt}$) exhibits a rapid increase early in training, quickly reaching near-optimal values (Figure 4a). This indicates the model's swift adoption of the required tripartite report structure. Concurrently, the Accuracy Reward ($R_{acc}$), encompassing both entity matching and uncertainty alignment, demonstrates a steady, more gradual improvement throughout the training process (Figure 4b). This reflects the nuanced challenge of enhancing medical accuracy and appropriate uncertainty expression in the "Findings" and "Impression" sections. Most notably, the Thinking Coherence Reward ($R_{coh}$) initially remains low but undergoes a significant and steep ascent in later training stages, eventually plateauing at a high level (Figure 4c). This trajectory strongly suggests that the reinforcement learning phase, guided by $R_{coh}$, successfully teaches the model to generate a logically sound "thinking" process that effectively connects radiological findings to clinical impressions. Collectively, these trends validate the multi-component reward function and the RL approach in progressively refining the model towards generating structurally correct, clinically accurate, and coherently reasoned radiology reports.

**Case Study**     To qualitatively illustrate the model's advanced capabilities, we present an analysis of Case Study in Figure 5. The model proficiently generated a clinically coherent and structured report, comprising distinct "Findings," "Thinking," and "Impression" sections, adhering to the desired output

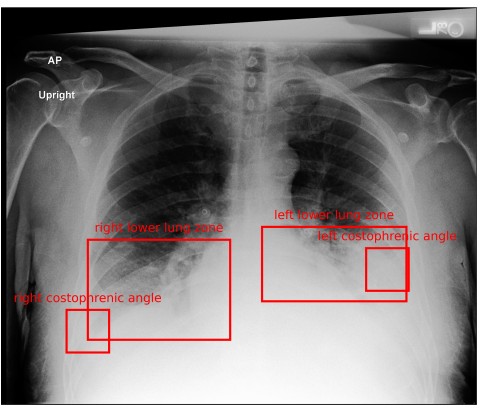

**Strong Correspondence in Key Findings:**

*Generated Findings: "...bibasal atelectasis... There is likely bilateral pleural effusions of moderate extent."*

*Ground Truth Findings: "GT: ...bibasilar opacities compatible with... adjacent atelectasis. Persistent moderate bilateral pleural effusions..."*

**Alignment in Overall Clinical Assessment:**

*Generated Impression: "...bibasal atelectasis and moderate pleural effusions... the concern is likely the effects of... management of any underlying atelectasis and effusions..."*

*Ground Truth Impression: "GT: Persistent moderate bilateral pleural effusions with adjacent atelectasis."*

(a) Chest X-ray image, the "bibasal atelectasis" mentioned in both findings and impression is corresponding to right and left lower lung zone as annotated in the image, and the "pleural effusions" is related with left and right costophrenic angle.

(b) Generated report sections.

Figure 5: Case Study: Illustrating robust VLM performance in identifying key clinical findings and generating a useful reasoning process. (b) Generated report sections demonstrate strong correspondence with ground truth on core observations such as bibasilar atelectasis and moderate pleural effusions, including appropriate use of uncertainty terms (e.g., "likely"). The model's explicit "<thinking>" section (analyzed in the main text, full content at D) provides a valuable, transparent pathway by linking these findings to patient context (e.g., supine position, HF history) and potential implications, showcasing the utility of structured reasoning in enhancing clinical report generation.

format. Due to page limit, key excerpts are shown, the full case is detailed in Appendix D. In the Findings section, the model accurately identified significant pathologies, including bibasilar atelectasis and moderate bilateral pleural effusions. These observations demonstrated strong correspondence with the ground truth report, and the model appropriately applied structural uncertainty terms (e.g., "likely"), showcasing its uncertainty-aware learning. The "Thinking" section proved particularly valuable, offering a transparent reasoning pathway. It successfully linked the identified visual findings with crucial patient context, such as supine positioning and history of biventricular heart failure, and considered their clinical implications. This explicit articulation of the thought process significantly enhances report interpretability and trustworthiness. Finally, the "Impression" section aligned well with the ground truth's primary assessment, correctly summarizing the key findings of atelectasis and effusions and focusing on their management. This case exemplifies the model's robust ability to produce accurate, contextually-aware, and clinically useful radiological interpretations with a clear and valuable reasoning structure, highlighting the strengths of our proposed framework.

## 5   Conclusion

We introduced CURV, a novel framework that enhances radiology report generation by integrating uncertainty awareness and explicit reasoning into vision-language models. CURV's key innovations—uncertainty modeling, a structured reasoning module, and a coherence-driven reinforcement learning strategy—enable the generation of reports that are clinically accurate, transparent, and appropriately express diagnostic confidence. Experimental results demonstrate CURV's superior performance in producing interpretable AI-generated CXR reports. However, CURV's performance relies on the quality of the initial curated datasets for uncertainty and reasoning, and its "thinking" process, while explicit, is learned via LLM-generated data. Furthermore, its generalization to other medical imaging modalities requires further investigation. Ultimately, the most critical next step is a large-scale clinical validation by expert radiologists to ensure its safe and effective translation into practice. Despite these limitations, CURV marks a significant advancement for trustworthy vision-language models in high-stakes clinical applications. To further promote research and transparency in this domain, the TRACE-CXR dataset developed for this study will be made publicly available. We believe CURV and the TRACE-CXR dataset will serve as valuable resources for future work where clarity in reasoning and uncertainty is crucial.

## Acknowledgments and Disclosure of Funding

This work is partially supported by the National Natural Science Foundation of China (62302413), the Health and Medical Research Fund (23220312), the General Research Fund RGC/HKBU12202621 from the Research Grant Council, and the Research Matching Grant Scheme RMGS2021_8_06 from the Hong Kong Government.

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

# A Experimental Setup

In this section, we outline the experimental setup for training and evaluating the CURV framework for radiology report generation and the comparison methods.

**Training Configuration**   Experiments were conducted on a node with 4xA100-80G GPUs. We utilized the Qwen-2.5-VL-3B as the backbone model for its balance of efficiency and performance. The training was configured with a batch size of 16, and a group size of 8 for GRPO in the reinforcement learning stage. The entire process spanned 3,200 steps over approximately 100 hours, with a learning rate of $1 \times 10^{-6}$ used for stable convergence.

To manage the memory requirements of training a 3B parameter model, we employed different fine-tuning strategies and state-of-the-art memory optimization techniques across the stages:

- **Stage 1 and 2 (SFT - Uncertainty Aware and Reasoning Initialization)**: To maximize efficiency, we kept the vision encoder frozen and fine-tuned only the parameters of the multimodal projection layer and the LLM backbone. This stage took approximately 4 hours to complete on 4 GPUs.

- **Stage 3 (RL - Enhancement)**: In this final stage, the goal was to refine the entire model's behavior based on global reward signals. Therefore, we performed full-parameter fine-tuning of the model checkpoint from Stage 2. To make this computationally intensive step feasible within the 80GB memory of each GPU, we leveraged a combination of standard optimization techniques:

    - **Mixed-Precision Training**: We used bfloat16 (bf16) precision, which halves the memory footprint for model parameters, gradients, and optimizer states compared to standard FP32, with minimal impact on training stability.
    - **Parameter and Optimizer State Sharding**: We utilized DeepSpeed with ZeRO Stage 3 optimization. This powerful technique partitions not only the optimizer states but also the gradients and the model parameters themselves across all 4 GPUs. This drastically reduces the per-GPU memory load, as each GPU only holds a fraction of the total training-related tensors.
    - **Activation Checkpointing**: To manage memory consumed by activations, we employed activation checkpointing (also known as gradient checkpointing). This technique avoids storing all activations by re-computing them during the backward pass, trading a modest amount of compute time for a significant reduction in memory usage.

**Baseline Models**   For the baseline models, CURV was benchmarked against established vision-language models, briefly introduced below with their experimental configurations:

- **LLaVA-1.5-7B** [21]: A vision language model for image and language understanding, fine-tuned on Vicuna with GPT-generated data for strong chat capabilities. We used the original model and a variant, LLaVA-1.5-7B-SFT-CXR, fine-tuned on a chest X-ray (CXR) dataset for 1 epoch.
- **MAIRA-2** [3]: A vision language model for radiology report generation from chest X-rays, we used the original model without additional fine-tuning.
- **HuatuoGPT-Vision-7B** [5]: A medical vision language model based on Qwen2-7B and LLaVA-v1.5, trained on PubMedVision for medical vision-language tasks. The original model was used without further fine-tuning.

All baselines were evaluated under consistent conditions, aligning with dataset and metric details in Sections 4.2 and 4.3, to ensure a fair comparison with CURV for chest X-ray report generation.

# B Datasets

We details the datasets and preprocessing steps undertaken to train and evaluate the CURV framework. A key contribution of our work is the curation and enhancement of existing datasets to specifically support uncertainty modeling and structured reasoning in radiology report generation.

Table 5: Dataset Statistics for CURV Training and Evaluation

| Statistic | Value |
|---|---|
| **MIMIC-CXR (Original)** | |
| Total Reports | 227,835 |
| Reports with Findings (REGEX) | 65.7% |
| Reports with Impression (REGEX) | 82.3% |
| Reports with Both Findings & Impression (REGEX) | 48.0% |
| **MIMIC-CXR (After LLM-Enhanced Parsing)** | |
| Reports with Findings | 185,122 (81.25%) |
| Reports with Impression | 193,755 (85.04%) |
| Reports with Both Findings & Impression | 151,048 (66.30%) |
| **Uncertainty Dataset ($\mathcal{D}_{\textbf{uncertainty}}$)** | |
| Samples with Uncertainty Annotations | 112,111 |
| Unique Uncertainty Expressions (freq. $> 5$) | ~2,700 |
| **TRACE-CXR Dataset ($\mathcal{D}_{\textbf{reason}}$)** | |
| Number of Reports in **TRACE-CXR** | 2000 |

**Core Dataset and LLM-Enhanced Parsing**   Our primary dataset is MIMIC-CXR [16, 17, 10], a large-scale collection of 227,835 radiology reports. Initial analysis using REGEX-based parsing revealed considerable heterogeneity in report structure. While "IMPRESSION" sections were present in 82.3% of reports and "FINDINGS" in 65.7%, only 48.0% of reports reliably contained both. This structural variability, coupled with an average raw text length of 634.3 characters (Findings: 335.2 chars, Impression: 175.1 chars), hindered consistent extraction of these crucial sections. To address these limitations and create a more uniform dataset for subsequent annotation and training, we employed a Large Language Model (LLM) for enhanced parsing of the MIMIC-CXR reports. This step significantly improved the availability of structured data: the proportion of reports with an identifiable "FINDINGS" section increased to 81.25% (185,122 reports), and those with an "IMPRESSION" section rose to 85.04% (193,755 reports). Consequently, the number of reports containing both "FINDINGS" and "IMPRESSION" sections increased substantially from 48.0% to 66.30% (151,048 reports), providing a more robust foundation for our work.

**Curating Data for Uncertainty Modeling**   A core innovation of CURV is its explicit modeling of diagnostic uncertainty. To train this capability (Stage 1 of our pipeline), we created a specialized uncertainty-annotated dataset, $D_{uncertainty}$. This was achieved by leveraging the Imagenome dataset [33], which provides valuable links between textual phrases in radiology reports and corresponding bounding box localizations on CXR images. Building upon these existing spatial annotations from Imagenome, we utilized the Qwen2.5-7B-instruct model to perform fine-grained uncertainty extraction. This model was specifically prompted to identify and extract uncertainty-expressing phrases directly from the report sentences that Imagenome had linked to specific visual findings. This process allowed us to map linguistic expressions of uncertainty to precise image regions. Our final uncertainty-annotated dataset contains 112,111 samples. Through this process, we identified approximately 2,700 unique uncertainty expressions that occurred with a frequency greater than five. The most common expressions included "likely" (131,334 instances), "may" (91,206 instances), and "could" (71,267 instances). For training, these annotations were formatted into structured JSON outputs, each containing the bounding box coordinates, an anatomical label, and the specific uncertainty description related to that finding (e.g., `{"bbox_2d": [121, 104, 180, 162], "label": "left lower lung zone", "uncertainty":"Bilateral nodular opacities that most likely represent nipple shadows."}`).

**Generating the TRACE-CXR Dataset ($\mathcal{D}_{\textbf{reason}}$) for Reasoning Initialization**   To initialize the basic reasoning capabilities in Stage 2 of the CURV framework, we constructed the **TRACE-CXR** dataset (also referred to as $\mathcal{D}_{\text{reason}}$ in our methodological descriptions in Section 3). The goal was to create training instances that explicitly model a "thinking" process connecting radiological findings to clinical impressions. The exact prompt used to guide the LLM in emulating an experienced radiologist's logical, step-by-step reasoning process is shown in Figure 6. Additionally, we validated the generated data using a separate prompt to ensure quality, retaining only those entries with a score higher than 80 out of 100, as shown in 7. Using the LLM-enhanced "FINDINGS" and "IMPRES-

```
You are a highly experienced radiologist tasked with generating a
    detailed reasoning process that explains how specific findings in
    a radiology report lead to a clinical impression. Your goal is to
    create a logical, step-by-step explanation that connects the
    observed features in a medical image to the final diagnosis or
    clinical takeaway. Use precise medical terminology and ensure the
    reasoning is clear, concise, and relevant to the provided
    findings and impression.
Input:
- Findings: [content of findings]
- Impression: [content of impression]
Task:
1. Analyze the provided findings and impression.
2. Generate a detailed reasoning process that explains how the
    findings support the impression. Break down the explanation into
    logical steps, addressing:
  - What specific abnormalities or features in the findings are most
      relevant to the impression.
  - How these features are typically associated with the suspected
      condition or diagnosis.
  - Any differential diagnoses or alternative possibilities
      considered based on the findings, and why the given impression
      is the most likely.
  - If applicable, mention any additional context (e.g., typical
      clinical presentations, risk factors, or imaging
      characteristics) that supports the reasoning.
3. Enclose your reasoning process in <thinking> tags as follows:
    <thinking>Your detailed reasoning here</thinking>.
Output Format:
<thinking>
[Your detailed step-by-step reasoning connecting the findings to the
    impression]
</thinking>
```

Figure 6: The LLM prompt for generating the 'thinking' section in the TRACE-CXR dataset. This prompt guides the LLM to create a logical, step-by-step reasoning process that connects the provided findings to the clinical impression, emulating a radiologist's thought process.

SION" sections as inputs, we prompted an LLM (grok-3) to generate an intermediate "THINKING" section. The LLM was guided by detailed instructions to emulate an experienced radiologist, tasking it to analyze the provided findings and impression, and then to construct a logical, step-by-step explanation of how the findings support the impression, including consideration of differential diagnoses where appropriate. The resulting data for $\mathcal{D}_{\text{reason}}$ consists of the CXR image paired with a structured report containing three distinct sections: <findings>Detailed description of image observations</findings>, <thinking>Reasoning based on findings</thinking>, and <impression>Concise summary and recommendations</impression>. Table 5 summarizes key statistics of the datasets pivotal to CURV's training and evaluation.

Table 6: Comprehensive comparison of evaluation scores between Clinicians (Clin), Grok3 (Grok), and Gemini 2.5 Pro (Gem).

| Study ID | Logical Coherence | | | Depth of Analysis | | | Relevance | | | Evidence Based | | | Consistency | | | Overall Score | |
|---|---|---|---|---|---|---|---|---|---|---|---|---|---|---|---|---|---|
| | Clin | Grok | Gem | Clin | Grok | Gem | Clin | Grok | Gem | Clin | Grok | Gem | Clin | Grok | Gem | Grok | Gem |
| s54517467 | 4 | 5 | 5 | 5 | 5 | 5 | 5 | 5 | 5 | 4 | 5 | 5 | 4 | 5 | 5 | 5.0 | 5.0 |
| s51966501 | 4 | 5 | 5 | 5 | 5 | 5 | 4 | 5 | 5 | 4 | 5 | 5 | 4 | 5 | 5 | 5.0 | 5.0 |
| s52188295 | 5 | 5 | 5 | 5 | 5 | 5 | 4 | 5 | 5 | 4 | 5 | 5 | 5 | 5 | 5 | 5.0 | 5.0 |
| s55493024 | 4 | 5 | 5 | 5 | 5 | 5 | 4 | 5 | 5 | 4 | 5 | 5 | 5 | 5 | 5 | 5.0 | 5.0 |
| s51074196 | 4 | 5 | 5 | 5 | 5 | 5 | 4 | 5 | 5 | 4 | 5 | 5 | 5 | 5 | 5 | 5.0 | 5.0 |
| s56689492 | 5 | 5 | 5 | 5 | 5 | 5 | 5 | 5 | 5 | 5 | 5 | 5 | 5 | 5 | 5 | 5.0 | 5.0 |
| s57002637 | 5 | 5 | 5 | 5 | 5 | 5 | 5 | 5 | 5 | 5 | 5 | 5 | 5 | 5 | 5 | 5.0 | 5.0 |
| s50849849 | 5 | 5 | 5 | 5 | 5 | 5 | 5 | 4 | 5 | 5 | 4 | 5 | 5 | 3 | 4 | 4.2 | 4.8 |
| s51215354 | 5 | 5 | 5 | 5 | 5 | 5 | 5 | 5 | 5 | 5 | 5 | 5 | 5 | 5 | 5 | 5.0 | 5.0 |
| s52528325 | 4 | 5 | 5 | 5 | 5 | 5 | 4 | 5 | 5 | 4 | 5 | 5 | 5 | 5 | 5 | 5.0 | 5.0 |

**Clinical Expert Evaluation of TRACE-CXR Dataset**   To validate the quality of the LLM-generated TRACE-CXR dataset and the reliability of our LLM-based evaluation protocol, we conducted a formal evaluation with a board-certified radiologist. We compared their expert judgments against those of two state-of-the-art LLMs (Grok3 and Gemini 2.5 Pro). For the evaluation, a random sample of 10 reports from the TRACE-CXR dataset was scored by all three evaluators using the exact same criteria outlined in Appendix C (Logical Coherence, Consistency, etc.). The results, summarized in Table 6, reveal a strong alignment between the human expert and both LLMs. Specifically, Gemini 2.5 Pro's scores were within one point of the clinician's 100% of the time, with a very low Mean Absolute Error (MAE) of 0.380. Similarly, Grok3's scores were within one point of the expert's 98.0% of the time, with an MAE of 0.440. This high degree of concordance provides strong, multi-faceted validation for our approach. It not only confirms that the LLM-generated reasoning traces in the TRACE-CXR dataset are of high clinical quality but also substantiates that our LLM-based evaluation metrics serve as a reliable proxy for expert human judgment.

```
**Task:** Evaluate a chest X-ray (CXR) report's <thinking> section
   across five key aspects.
**Instruction:** Assess the <thinking> section of a chest X-ray
   report across multiple aspects. Analyze how well the reasoning
   links findings to impressions, its depth, clinical relevance,
   evidence base, and consistency with other sections.
**Input Format:**
- '<findings>': Observations from the CXR image.
- '<thinking>': Reasoning or analysis based on findings.
- '<impression>': Summarized conclusions or diagnosis.
**Provided Input:**
{findings}{thinking}{impression}
**Evaluation Criteria:**
Evaluate ALL of the following aspects independently:
1. **Logical Coherence:**
   - **Score 5:** Clear, logical flow, seamlessly connecting findings
      to impressions without doubt.
   - **Score 0:** Incoherent or illogical; reasoning is fragmented or
      fails to connect findings to impressions.
2. **Depth of Analysis:**
   - **Score 5:** Deep analysis with comprehensive explanations,
      including alternatives or limitations.
   - **Score 0:** Superficial or absent analysis; merely restates
      findings without insight.
3. **Relevance:**
   - **Score 5:** Highly relevant, focusing on key clinical findings
      without extraneous content.
   - **Score 0:** Irrelevant or off-topic reasoning, ignoring
      clinical context.
4. **Evidence-based:**
   - **Score 5:** Strongly evidence-based, tied to medical knowledge
      and practices.
   - **Score 0:** Lacking evidence; speculative or contrary to
      medical standards.
5. **Consistency:**
   - **Score 5:** Fully consistent across '<findings>', '<thinking>',
      and '<impression>'.
   - **Score 0:** Grossly inconsistent with contradictions
      undermining trustworthiness.
**Output Format:** Return evaluation as a JSON object per the
   provided schema.
```

Figure 7: The LLM-based evaluation prompt for the 'Thinking' section. This prompt instructs the evaluator LLM to score the generated reasoning on five criteria—Logical Coherence, Depth of Analysis, Relevance, Evidence-Based Nature, and Consistency—and return the assessment in a structured JSON format.

```
**Task:** Comprehensively evaluate uncertainty expressions in a chest
    X-ray (CXR) report, comparing a generated report with a ground
    truth report.
**Instruction:** You are tasked with rigorously assessing how well
    the generated report expresses uncertainty compared to the ground
    truth report. Focus on two key aspects:
1. **Structural Uncertainty:** Hedging or ambiguity about specific
    anatomical regions or findings (typically in findings sections)
    - Example: "a nodule may represent a benign lesion or malignancy"
    - This appears in descriptions of specific observations
2. **Semantic Uncertainty:** Ambiguity in overall diagnostic
    synthesis (typically in impression sections)
    - Example: "findings are nonspecific and could be consistent with
        infection"
    - This appears in the overall assessment/conclusion
First determine which parts of each report represent findings vs.
    impression sections, even if they aren't explicitly labeled. Then
    compare the uncertainty expressions between generated and ground
    truth reports.
**Evaluation Criteria:**
1. **Structural Uncertainty:**
    - **Score 5:** Generated report contains uncertainty expressions
        about specific findings highly similar to ground truth in both
        content and strength
    - **Score 3:** Somewhat similar with noticeable differences
    - **Score 0:** No structural uncertainty or completely dissimilar
        to ground truth
2. **Semantic Uncertainty:**
    - **Score 5:** Generated report contains holistic diagnostic
        uncertainty expressions highly similar to ground truth
    - **Score 3:** Somewhat similar with noticeable differences
    - **Score 0:** No semantic uncertainty or completely dissimilar to
        ground truth
**Output Format:** Return your evaluation as a JSON object with two
    main sections (structural_uncertainty and semantic_uncertainty)
    each containing:
- score (0-5)
- explanation (1-2 sentences justifying the score)
- triples_comparison (comparison of uncertainty triples - subject,
    uncertainty term, interpretation)
- uncertainty_strength (comparison of strength of uncertainty terms)
- contextual_appropriateness (assessment of whether uncertainty is
    expressed in appropriate contexts)
Also include an overall_score section with the same fields (calculate
    score as average of the two aspects).
**Provided Input:**
Generated Report:
{generated_text}
Ground Truth Report:
{ground_truth_text}
```

Figure 8: The LLM-based evaluation prompt for uncertainty expression. This prompt directs the evaluator LLM to assess and score the generated report's handling of both structural (finding-specific) and semantic (diagnostic) uncertainty in comparison to the ground truth report.

## C   Evaluation Metric

To comprehensively assess the performance of CURV and baseline methods, we employ a suite of evaluation metrics targeting clinical accuracy, the quality of uncertainty expression, and the coherence of the generated reasoning pathways.

**Clinical Accuracy and NLP Metrics**    For evaluating the factual correctness and fluency of the generated "Findings" and "Impression" sections, we utilize standard Natural Language Processing (NLP) metrics, including BLEU (1-4) [24], ROUGE-L [19], METEOR [2]. Beyond these, we measure clinical accuracy using F1-scores based on medical entity extraction. Specifically, we employ CheXbert [30] and Radgraph [13] to calculate the f1 score of extract entities.

**LLM-based Evaluation**    To evaluate the "Thinking" section generated by CURV, which is not present in standard reports and thus not amenable to traditional reference-based metrics, we leverage an LLM-based evaluation protocol. We prompt a capable LLM (Qwen3-32B [36]) to assess the quality of the "Thinking" section based on five crucial aspects: Logical Coherence, Depth of Analysis, Relevance, Evidence-Based Nature, and Consistency with the "Findings" and "Impression" sections. And we also evaluate the expression of uncertainty, particularly distinguishing between structural and semantic uncertainty, also requires a more sophisticated approach than standard metrics. We employ another LLM-based evaluation protocol for this purpose. The LLM is tasked with comparing the uncertainty expressions in a generated report against those in a ground truth report, focusing on two key aspects: Structural Uncertainty and Semantic Uncertainty. The prompt for the LLM-based evaluation is shown in Figure 7 and 8.

## D   Full Case Study

This appendix provides the complete details to case study as shown in Figure 9.

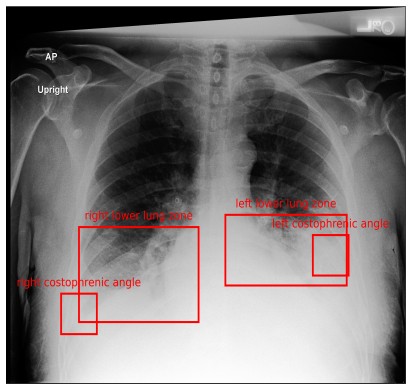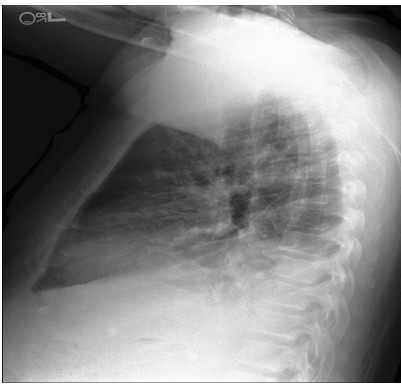

(a) Frontal and Lateral Images

**Ground Truth: Findings and Impression**

**FINDINGS:** There are persistent bibasilar opacities compatible with layering effusions with adjacent atelectasis and perhaps mild pulmonary edema. The cardiomediastinal silhouette is stable. Atherosclerotic calcifications seen at the aortic arch. No acute osseous abnormalities.

**IMPRESSION:** Persistent moderate bilateral pleural effusions with adjacent atelectasis. Superimposed infection would be difficult to exclude.

**Model Output: Findings, Thinking and Impression**

**FINDINGS:** He has bibasal atelectasis, likely related to poor mobilization of lung volumes during inspiration. There is likely bilateral pleural effusions of moderate extent. However, there are no pulmonary parenchymal opacities at the apices; the pleural surface appears clear. Given patient's history of biventricular HF, she is supine on the exam, resulting in her moderate and greater than moderate apical lung collapse. No evidence of new cardiomegaly or acute pulmonary vascular congestion.

**THINKING:** Since the imaging shows bibasal atelectasis, a clear pleural surface, and no new signs of cardiomegaly or pulmonary vascular congestion, the history of biventricular heart failure likely plays a significant role. The patient is described as supine on the exam, which contributes to the bibasal atelectasis but does not suggest acute worsening of the pulmonary condition. The presence of moderate pleural effusions is noted, and their lack of apical opacities could indicate no underlying pulmonary parenchymal disease such as pneumonia. Therefore, the primary concern is likely the presence and management of the atelectasis and effusions, given the patient's cardiopulmonary status.

**IMPRESSION:** The patient is supine and therefore has bibasal atelectasis and moderate pleural effusions. There is no evidence of pulmonary parenchymal opacities at the apices, and the pleural surface is clear. The presence of moderate pleural effusions suggests that the lung volumes were better mobilized during expiration. With patient's history of biventricular HF, the concern is likely the effects of the supine positioning on lung volumes and the management of any underlying atelectasis and effusions, though no acute pulmonary or cardiac changes are seen.

Figure 9: A full, detailed view of the case study. This figure presents the frontal and lateral CXR images, the ground truth report, and the complete, three-part report generated by CURV, showcasing the model's ability to produce a clinically coherent and well-structured output.

# E    Licenses for External Assets

This appendix details the licenses for publicly available datasets and pre-trained models utilized in this work.

**Datasets**

- **MIMIC-CXR (version 2.1.0):**
  - **Source:** Johnson et al. [17].
  - **URL:** `https://physionet.org/content/mimic-cxr/2.1.0/`
  - **License:** PhysioNet Credentialed Health Data License 1.5.0. Access requires credentialing and signing a data use agreement.

- **Chest ImaGenome Dataset (version 1.0.0):**
    - **Source:** Wu et al. [33].
    - **URL:** `https://physionet.org/content/chest-imagenome/1.0.0/`
    - **License:** Derived from MIMIC-CXR, subject to the PhysioNet Credentialed Health Data License 1.5.0.

**Pre-trained Models and Baselines**

- **Qwen-2.5-VL-3B (Backbone for CURV):**
    - **Source:** Yang et al. [36].
    - **URL:** `https://github.com/QwenLM/Qwen3`
    - **License:** Apache 2.0 License
- **LLaVA-1.5-7B:**
    - **Source:** Liu et al. [21].
    - **URL:** `https://github.com/haotian-liu/LLaVA`
    - **License:** Apache 2.0 License.
- **MAIRA-2:**
    - **Source:** Bannur et al. [3].
    - **URL:** `https://huggingface.co/microsoft/maira-2`
    - **License:** MICROSOFT RESEARCH LICENSE TERMS
- **HuatuoGPT-Vision-7B:**
    - **Source:** Chen et al. [5].
    - **URL:** `https://huggingface.co/FreedomIntelligence/HuatuoGPT-Vision-7B`
    - **License:** Apache 2.0 License
- **RadGraph:**
    - **Source:** Jain et al. [13].
    - **URL:** `https://huggingface.co/StanfordAIMI/RRG_scorers`
    - **License:** MIT License.
- **CheXbert:**
    - **Source:** Smit et al. [30].
    - **URL:** `https://huggingface.co/StanfordAIMI/RRG_scorers`
    - **License:** MIT License.

