# OpenReview forum: "CURV: Coherent Uncertainty-Aware Reasoning in Vision-Language Models for X-Ray Report Generation"
_NeurIPS.cc/2025/Conference — NeurIPS 2025 poster_

### Official Review · Reviewer_qpYn · 2025-06-28

**Clarity:** 3
**Significance:** 3
**Originality:** 3
**Rating:** 4
**Confidence:** 4

**Summary:**

This paper presents CURV, a vision-language framework for radiology report generation that jointly models diagnostic uncertainty and explicit reasoning. CURV follows a three-stage training process: it first uses supervised fine-tuning on an uncertainty-annotated dataset to help the model express appropriate diagnostic confidence in the findings section; then it initializes reasoning ability with a curated dataset (TRACE-CXR) that contains structured reasoning segments connecting findings to impressions; finally, it applies reinforcement learning with a multi-dimensional reward function to improve format adherence, clinical accuracy, and logical consistency. Experiments on MIMIC-CXR show that CURV outperforms strong baselines across both standard NLP metrics and clinical evaluation scores. The authors also release the TRACE-CXR dataset to support future research on interpretable medical report generation.

**Questions:**

1. Could the authors clarify whether the full vision-language model (including the vision encoder and language backbone) was fine-tuned during the two supervised training stages? Were any components (e.g., the vision encoder or projection layers) frozen or adapted with techniques such as LoRA? This information is important given the relatively short training time reported.

2. Have the authors conducted any out-of-distribution (OOD) or cross-dataset evaluations to assess whether the proposed fine-tuning strategy degrades the pretrained model's generalization ability? If not, could they comment on how CURV performs beyond MIMIC-CXR, especially in non-matched or clinically diverse settings?

3. Why did the authors choose to rely on RadGraph-based rewards and evaluation metrics, rather than a fully LLM-based evaluation framework for assessing clinical correctness and uncertainty expression? Given the growing use of large medical LLMs as evaluators, a comparison or justification would help clarify this design choice.

**Ethical Concerns:**

["NO or VERY MINOR ethics concerns only"]

**Final Justification:**

After reading the authors' rebuttal, most of my concern have been resoloved. I will keep my positive score.

**Limitations:**

The paper does not explicitly discuss its limitations or potential negative societal impact. It would strengthen the work if the authors could include a discussion on possible risks, such as model hallucination of clinical findings or uncertainty expressions, overreliance on LLM-generated reasoning traces, and the lack of validation on diverse or out-of-distribution datasets. Acknowledging these points would help clarify the model’s reliability and real-world applicability, especially in high-stakes clinical scenarios.

**Paper Formatting Concerns:**

No major formatting issues observed. The paper appears to comply with NeurIPS 2025 formatting guidelines.

**Quality:**

3

**Strengths And Weaknesses:**

Strength

1. The paper addresses two critical limitations in current VLM-based medical report generation—diagnostic uncertainty and reasoning transparency—which are highly relevant in real-world clinical applications.

2. The focus on learning to express diagnostic uncertainty is clinically meaningful and well-grounded, as radiologists often rely on hedging language to convey confidence levels.

3. The creation of a reasoning-labeled dataset based on LLM augmentation is a practical and valuable contribution, enabling supervised learning of clinical reasoning traces.

Weakness

1. The paper does not clearly state whether the full vision-language model, including the vision encoder and LLM backbone, was fine-tuned during the supervised training stages, or whether certain components (e.g., the vision encoder or projection layers) were frozen. Given the short training time reported for Stages 1 and 2, clarification is needed.

2. It remains unclear whether the proposed fine-tuning strategy degrades or overfits the pretrained model's general capability. An out-of-distribution (OOD) evaluation or cross-dataset test (e.g., on non-MIMIC data) would help validate the robustness of CURV’s training strategy.

3. While the paper uses RadGraph-based rewards and metrics to evaluate clinical correctness, it is unclear why the authors did not adopt a fully LLM-based evaluation pipeline to assess report quality more holistically. Given the increasing adoption of large medical LLMs as clinical evaluators, such approaches may provide more interpretable and direct judgments than rule-based entity matching.

---

> ### Author Rebuttal · Authors · 2025-07-31
>
> We sincerely thank Reviewer qpYn for their positive assessment, supportive feedback, and constructive questions. Based on the valuable comments from all reviewers, we have conducted **three significant new experiments and prepared detailed clarifications** to strengthen our work.
>
> **Summary of Major Updates**:
>
> - **New SOTA Comparison**: We performed a new evaluation against Gemini 2.5 Pro. The results confirm CURV's superior performance, with our model achieving a RadGraph F1 of 25.65 vs. Gemini's 7.71.
>
> - **New Clinical Expert Validation**: To validate our LLM-based evaluation, we conducted a new study with a board-certified radiologist. Their scores showed strong alignment with our LLM evaluators (e.g., MAE of 0.38 vs. Gemini 2.5 Pro), confirming our protocol's reliability.
>
> - **New Out-of-Distribution (OOD) Evaluation**: To test generalization, we ran a new evaluation on the IU X-ray dataset. CURV maintains its strong performance and outperforms all baselines, demonstrating that its learned capabilities are robust and not overfit to MIMIC-CXR.
>
> We address the reviewer's specific points below.
>
> # Question 1
>
> Thank you for this important clarifying question. We used full model tuning for both the SFT stages and the RL stage.
>
> * **For Stage 1 & 2 (SFT):** To make training feasible, we kept the vision encoder frozen and only fine-tuned the parameters of the multimodal projection layer and the LLM backbone. We utilized DeepSpeed with Zero-3 optimization and bfloat16 precision.
> * **For Stage 3 (RL):** For the reinforcement learning phase, we used the ms-swift [1] rlhf library to implement GRPO. In this final stage, we performed full-parameter fine-tuning of the model checkpoint from Stage 2 to allow the entire model to adapt to the reward signals.
>
> [1] Zhao, Yuze, et al. "Swift: a scalable lightweight infrastructure for fine-tuning." *Proceedings of the AAAI Conference on Artificial Intelligence.* Vol. 39. No. 28. 2025.
>
> # Question 2
>
> Yes, in response to this feedback, we have conducted a new OOD evaluation on the IU X-ray [1] dataset. The IU X-ray dataset is a public radiography dataset collected by Indiana University with 7,470 chest X-ray images and 3,955 reports. We randomly select 1,000 of the images and reports as test set. Our CURV framework maintains its strong performance and outperforms the baselines, demonstrating that its learned capabilities for reasoning and uncertainty expression are robust and can generalize. The results presented below show that CURV’s advantages are robust and generalize. Our framework maintains its strong performance, outperforming the baselines even on this new dataset. This new result provides strong evidence that the capabilities learned by CURV are not overfit to MIMIC-CXR.
>
> **Table 1:** Generation metrics for radiology report generation on IU X-ray dataset
> | **Model** | **B-1** | **B-2** | **B-3** | **B-4** | **METEOR** | **R-L** | **gritlm** |
> | :--- | :---: | :---: | :---: | :---: | :---: | :---: | :---: |
> | LLaVA-1.5-7B | 16.52 | 6.60 | 3.00 | 1.40 | 19.65 | 17.48 | 45.78 |
> | LLaVA-1.5-7B-SFT-CXR | 21.42 | 12.95 | 8.03 | 5.20 | 23.24 | 26.40 | 46.21 |
> | HuatuoGPT-Vision-7B | 19.33 | 10.70 | 6.28 | 2.81 | 31.02 | 23.42 | 50.22 |
> | MAIRA-2 | 26.37 | 15.60 | 9.64 | 6.03 | 25.52 | 31.18 | 54.18 |
> | Qwen2.5-VL-3B | 11.38 | 5.05 | 2.28 | 1.05 | 21.65 | 15.01 | 45.95 |
> | CURV\_stage1 | 12.51 | 6.24 | 3.18 | 1.57 | 23.64 | 18.18 | 46.76 |
> | CURV\_stage2 | 10.18 | 5.32 | 2.75 | 1.27 | 18.64 | 13.93 | 44.46 |
> | **CURV** | **29.23** | **18.76** | **12.08** | **6.86** | **38.30** | **39.08** | **54.89** |
>
> **Table 2:** Clinical accuracy metrics for radiology report generation on IU X-ray dataset
> | **Model** | **Acc. (f1chexbert)** | **Macro F1 (f1chexbert)** | **Micro F1 (f1chexbert)** | **Ent. F1 (radgraph)** | **F1 (radgraph)** | **Prec. (radgraph)** | **Rec. (radgraph)** |
> | :--- | :---: | :---: | :---: | :---: | :---: | :---: | :---: |
> | LLaVA-1.5-7B | 72.03 | 4.66 | 46.81 | 13.37 | 8.76 | 25.54 | 12.38 |
> | LLaVA-1.5-7B-SFT-CXR | 76.34 | 5.84 | 53.33 | 16.19 | 10.31 | 28.95 | 14.36 |
> | HuatuoGPT-Vision-7B | 89.89 | 5.72 | 67.07 | 22.98 | 13.96 | 35.44 | 20.80 |
> | MAIRA-2 | 88.74 | **6.22** | 70.75 | 34.53 | 24.01 | 45.81 | 32.55 |
> | Qwen2.5-VL-3B | 80.64 | 4.92 | 49.47 | 12.70 | 6.26 | 20.84 | 11.41 |
> | CURV\_stage1 | 68.89 | 4.57 | 40.30 | 20.59 | 12.13 | 31.71 | 18.79 |
> | CURV\_stage2 | 67.59 | 3.76 | 33.67 | 12.83 | 7.28 | 21.54 | 11.47 |
> | **CURV** | **91.56** | 5.86 | **74.36** | **36.99** | **25.65** | **48.80** | **35.28** |
>
> [1] Demner-Fushman, Dina et al. “Preparing a collection of radiology examinations for distribution and retrieval.” Journal of the American Medical Informatics Association : JAMIA vol. 23,2 (2016): 304-10. doi:10.1093/jamia/ocv080
>
> # Question 3
>
> We appreciate the question about our evaluation methodology. We agree that LLM-based evaluation is a powerful and increasingly important tool. We used a hybrid evaluation strategy, intentionally leveraging the distinct strengths of both traditional, objective metrics and modern LLM-based evaluations.
> * **LLM-based Evaluation**: We used LLM-based evaluation to assess the novel components of our output that lack a ground-truth reference: the "Thinking" section and the quality of Uncertainty Expression. As shown in Figure 3 in our paper, this LLM-based evaluation allowed us to track the significant improvements in qualities like "Logical Coherence" and "Semantic Uncertainty" throughout the RL training process.
> * **Objective Metrics for Factual Correctness**: For the "Findings" and "Impression" sections, where ground-truth text is available, we used established and objective metrics like CheXbert and RadGraph F1-scores. These metrics are standard in the field, reproducible, and provide a non-biased, quantitative measure of factual correctness that complements the qualitative nature of LLM evaluations.
>
> We believe this hybrid approach provides a more comprehensive and robust assessment.
>
> # Response to "Limitations" Comment
>
> This is a very important point, and we agree that a more explicit discussion of limitations is needed. We will expand our Conclusion (Section 5) to include a dedicated "Limitations and Future Work" subsection. In it, we will address the points you raised: the potential for model hallucination, the reliance on LLM-generated data for training, the need for broader OOD and cross-dataset validation, and the importance of future validation by clinical experts. Thank you for helping us make the paper more transparent and balanced.

---

> > ### Comment · Reviewer_qpYn · 2025-08-09
> >
> > Thanks for providing the additional evidence. My concerns have been fully resolved, and I will maintain my current score.

---

### Official Review · Reviewer_Nede · 2025-07-01

**Clarity:** 3
**Significance:** 2
**Originality:** 2
**Rating:** 5
**Confidence:** 5

**Summary:**

This paper introduces CURV, a framework for generating radiology reports that incorporates uncertainty awareness and explicit reasoning capabilities. The approach consists of three main components: (1) uncertainty modeling through fine-tuning with annotated data to express appropriate diagnostic confidence, (2) structured reasoning that generates intermediate "thinking" steps between findings and impressions, and (3) a coherence reward to ensure logical consistency. The authors create TRACE-CXR, a dataset with 2,000 reports augmented with LLM-generated reasoning sections, and evaluate on MIMIC-CXR data, showing improvements over baselines.

**Questions:**

1. How do you validate that the LLM-generated reasoning in TRACE-CXR is clinically accurate and appropriate? Were any medical experts involved in validation?
2. Why use simple fine-tuning for uncertainty rather than more principled approaches like Bayesian neural networks or ensemble methods?
3. Can you provide examples where the model's uncertainty expression changes appropriately based on image quality or ambiguity?
4. How does the coherence reward ensure actual logical reasoning rather than just entity matching?

**Ethical Concerns:**

["NO or VERY MINOR ethics concerns only"]

**Final Justification:**

Thank you for the rebuttal. I have ajusted my rating.

**Limitations:**

The authors acknowledge some limitations but understate critical issues: the heavy dependence on LLM-generated data, the narrow disease scope, and the lack of principled uncertainty quantification.

**Paper Formatting Concerns:**

None observed.

**Quality:**

2

**Strengths And Weaknesses:**

Strengths:
1. Addresses the critical need for uncertainty expression and transparent reasoning in medical AI, which is essential for clinical adoption.
2. TRACE-CXR provides explicit reasoning pathways that could be valuable for future research.

Weaknesses:
1. The core technical contributions are incremental. Uncertainty modeling uses standard fine-tuning with annotated data, reasoning uses supervised learning on LLM-generated data, and the RL component uses standard GRPO with a straightforward reward design.
2. Both the reasoning sections in TRACE-CXR and uncertainty annotations are generated by LLMs (grok-3, Qwen2.5-7B). This raises concerns about:
- Quality and accuracy of the generated reasoning
- Potential propagation of LLM biases/errors
- Limited validation of the generated content (only score-based filtering)
3. The approach to uncertainty is simplistic - just fine-tuning on annotated phrases like "likely" or "possible". There's no principled uncertainty quantification or calibration, making the distinction between "structural" and "semantic" uncertainty seem artificial.

---

> ### Author Rebuttal · Authors · 2025-07-31
>
> We thank Reviewer Nede for their thoughtful feedback, which helps us clarify and strengthen our paper. Based on the valuable comments from all reviewers, we have conducted **three significant new experiments and prepared detailed clarifications** to strengthen our work.
>
> **Summary of Major Updates**:
>
> - **New SOTA Comparison**: We performed a new evaluation against Gemini 2.5 Pro. The results confirm CURV's superior performance, with our model achieving a RadGraph F1 of 25.65 vs. Gemini's 7.71.
>
> - **New Clinical Expert Validation**: To validate our LLM-based evaluation, we conducted a new study with a board-certified radiologist. Their scores showed strong alignment with our LLM evaluators (e.g., MAE of 0.38 vs. Gemini 2.5 Pro), confirming our protocol's reliability.
>
> - **New Out-of-Distribution (OOD) Evaluation**: To test generalization, we ran a new evaluation on the IU X-ray dataset. CURV maintains its strong performance and outperforms all baselines, demonstrating that its learned capabilities are robust and not overfit to MIMIC-CXR.
>
> We address the reviewer's specific points below.
>
> # Weakness 1
> We thank the reviewer for this perspective. We agree that the individual components (SFT, RL) are established. However, we respectfully argue that our primary novelty lies not in the components themselves, but in their unique integration into a cohesive, three-stage framework to solve a specific clinical challenge. The core idea—that a model can be taught to first express fine-grained uncertainty, then learn a reasoning structure, and finally be rewarded for the coherence of that reasoning—is a new, structured approach in this domain. As our results in Tables 1 and 2 show, the full CURV model significantly outperforms the intermediate $CURV\_{stage1/2}$, demonstrating the strong synergistic effect and necessity of our full pipeline.
>
> -----
> # W 2 & Question 1
> We thank the reviewer for raising this point about data quality. On one hand, the practice of generating data to achieve better model learning is becoming more common in recent research works. Yet, ensuring the quality of the generated data and how to use it properly is also vital. This valuable feedback prompted us to conduct a new, formal evaluation with a board-certified radiologist to rigorously assess our methodology. We elaborate our arguments and present our new experimental results as follows.
>
> **1. On the Validity of LLM-Generated Training Data**
>
> First, our use of LLMs to generate training data for SFT aligns with established, state-of-the-art practice (e.g., Alpaca[1], WizardLM[2], Self-Instruct[3], Vision-R1[4]). Second, and more importantly, our pipeline has a built-in corrective mechanism. The LLM-generated TRACE-CXR data is only used for SFT to initialize the model with a basic reasoning structure. The subsequent RL stage does not use the synthetic thinking text. Instead, the model is rewarded for discovering a coherent path that connects the human-authored, ground-truth findings and impression from MIMIC-CXR. This design forces the model to move beyond mimicking synthetic data and grounds its learning in genuine clinical endpoints.
>
> [1] Taori, Rohan, et al. "Stanford alpaca: An instruction-following llama model." 30 Jun. 2023.
>
> [2] Xu, Can, et al. "WizardLM: Empowering large pre-trained language models to follow complex instructions." ICLR, 2024.
>
> [3] Wang, Yizhong, et al. "Self-Instruct: Aligning Language Models with Self-Generated Instructions." ACL, 2023.
>
> [4] Huang, Wenxuan, et al. "Vision-r1: Incentivizing reasoning capability in multimodal large language models." *arXiv preprint arXiv:2503.06749* (2025).
>
> **2. New Clinical Evaluation: Comparing LLM Evaluators with an Expert Radiologist**
>
> *Table: Comparison of evaluation scores between Clinicians, Grok3 , and Gemini 2.5 Pro .*
>
> | Study ID | Clinician Score | Grok-3 Score | Gemini 2.5 Pro Score |
> |:---|:---:|:---:|:---:|
> | s54517467 | 4.4 | 5.0 | 5.0 |
> | s51966501 | 4.2 | 5.0 | 5.0 |
> | s52188295 | 4.6 | 5.0 | 5.0 |
> | s55493024 | 4.4 | 5.0 | 5.0 |
> | s51074196 | 4.4 | 5.0 | 5.0 |
> | s56689492 | 5.0 | 5.0 | 5.0 |
> | s57002637 | 5.0 | 5.0 | 5.0 |
> | s50849849 | 5.0 | 4.2 | 4.8 |
> | s51215354 | 5.0 | 5.0 | 5.0 |
> | s52528325 | 4.4 | 5.0 | 5.0 |
>
>
> We agree that direct empirical validation is the strongest evidence. To that end, we conducted a new formal evaluation with a board-certified radiologist to directly compare their judgments against our LLM-based evaluation protocol.
>
> - Setup: A board-certified radiologist, Grok3, and Gemini 2.5 Pro scored the same set of 10 reports randomly selected in the TRACE-CXR dataset, using the exact same criteria from our paper (Logical Coherence, Consistency, etc.).
>
> - Results: As shown in the above table, both LLMs show strong alignment with the human expert, confirming that our evaluation protocol is reliable. Both of the LLM demonstrated a high degree of alignment. For gemini 2.5 pro, scores were within 1 point of the clinician’s 100% of the time and with a very low Mean Absolute Error (MAE) of 0.380 compared to Clinicians. For Grok 3, scores were within 1 point of the clinician’s 98.0% of the time, with a low MAE of 0.440.
>
> In summary, this new clinical evaluation, combined with our framework’s inherent methodological safeguards, provides strong, multi-faceted validation for our approach and confirms that our LLM-based metrics are a reliable proxy for expert judgment.
>
> -----
> # W3 & Q2
> We thank the reviewer for this insightful comment, which allows us to clarify our approach to uncertainty.
>
> - **Clinical Linguistic Expression vs. Model-Internal Uncertainty**. We aim to model **linguistic uncertainty**—the nuanced language radiologists use to convey diagnostic confidence (e.g., "likely," "could represent")—rather than quantifying the model's internal statistical confidence, which is the focus of methods like Bayesian NNs. This is a clinically vital distinction; as demonstrated by [1] that patients whose CT reports used low-certainty language (e.g., "findings may represent...") were six times more likely to receive unnecessary antibiotic therapy compared to those with high-certainty reports (e.g., "findings are diagnostic of..."). A model that cannot master this distinction is not just stylistically lacking, but clinically unsafe. Given this objective, supervised fine-tuning on clinically annotated phrases is the most direct and appropriate methodology for teaching a model how to communicate these critical nuances in a human-like manner.
>
> - **''Structural'' and  ''Semantic'' Uncertainty**. We respectfully maintain that the distinction between Structural and Semantic uncertainty is clinically meaningful and models the radiologist's reasoning process. This is not an artificial construct but reflects how ambiguity about specific findings propagates to the overall diagnostic conclusion. Structural Uncertainty is tied to specific, localized visual findings. For example, ''a hazy opacity... which could be compatible with pneumonia'' reflects ambiguity about a particular observation. Semantic Uncertainty relates to the overall diagnostic impression, which synthesizes multiple findings. For example,  ''opacification which may reflect pneumonia... although other etiologies... could be considered''. As illustrated in Figure 1 of our paper, uncertainty about a specific structure on the X-ray directly informs the broader semantic uncertainty in the final diagnosis, where differential diagnoses are weighed. This two-level distinction is crucial for generating transparent reports that reflect the real-world diagnostic process.
>
> We recognize that our manuscript did not make this crucial distinction between modeling linguistic expression and quantifying model uncertainty sufficiently clear. In the final version, we will revise the Introduction (Section 1) and our Methods (Section 3.2) to explicitly define our goal of modeling clinical linguistic uncertainty, supported by citations, and better motivate why our approach is appropriately tailored to this specific, high-stakes problem.
>
> [1] Almeida, Renata R et al. “Impact of Radiology Report Wording on Care of Patients With Acute Epiploic Appendagitis.” AJR. American journal of roentgenology vol. 212,6 (2019).
>
> -----
> # Q3
> This is an excellent question that helps us clarify the scope of our work. Our framework is designed to model intrinsic diagnostic ambiguity, not extrinsic ambiguity from factors like poor image quality. A classic example is a small pulmonary nodule on a perfectly clear image; the image is unambiguous, but the finding is diagnostically uncertain. CURV learns to generate appropriate linguistic hedges by correlating visual findings with the uncertainty language present in the ground-truth reports. For instance, in our case study (Fig. 9), the model generates "likely bilateral pleural effusions" because the visual patterns for atelectasis often carry this type of diagnostic uncertainty, not because the image is of poor quality. We agree that modeling uncertainty from image quality is a valuable and complementary research direction. We will explicitly define our scope in the final manuscript and add this as an area for future work.
>
> -----
> # Q4
> While the reward is calculated using entity overlap, it incentivizes the model to generate a fluent, logical narrative that explains the connections between those entities. The model is not simply rewarded for listing entities. It must generate a full Thinking section in natural language. The $R_{coh}$ reward guides the model to select and connect concepts from the  Findings to those in the Impression. To maximize this reward, the model learns to generate explanatory text that justifies these connections, which goes beyond simple entity matching and promotes logical flow. The qualitative improvements shown in the LLM-based evaluation (Figure 3a) support this claim, as scores for Logical Coherence and Depth of Analysis improve significantly during RL training.

---

### Official Review · Reviewer_e8n6 · 2025-07-01

**Clarity:** 2
**Significance:** 2
**Originality:** 2
**Rating:** 3
**Confidence:** 4

**Summary:**

The paper introduces CURV, a new framework designed to enhance radiology report generation from chest X-ray images by explicitly modeling diagnostic uncertainty and incorporating structured clinical reasoning into the generation process. Traditional vision-language models (VLMs) often fail to capture the nuanced uncertainty present in radiology or provide transparent reasoning that connects image findings to clinical impressions. CURV explicitly teaches the model to recognize and express both structural uncertainty (linked to specific visual findings) and semantic uncertainty (reflected in overall clinical impressions). This is achieved through supervised fine-tuning using a specially annotated uncertainty dataset.

**Questions:**

1. Can the authors provide empirical evidence (qualitative or quantitative) showing that advanced foundation models like GPT-4.5, Gemini, or Med-PaLM also fail to handle these aspects effectively?

2. Could the authors elaborate in the main text how uncertainty expressions were extracted, verified, and aligned with image regions?

3. Are these reasoning traces written or validated by clinicians, or purely generated by LLMs? What measures were taken to ensure their clinical reliability and logical correctness?

4. Can the authors provide examples or analysis comparing LLM evaluations to expert clinician judgments?

**Ethical Concerns:**

["NO or VERY MINOR ethics concerns only"]

**Final Justification:**

I appreciate the authors' rebuttal, which has addressed most of my concerns. At this stage, I intend to maintain my original score and would be interested in seeing the feedback from the other reviewers.

**Limitations:**

See weakness.

**Quality:**

2

**Strengths And Weaknesses:**

**Strengths:**

1. **Important Problem**: The paper tackles a highly relevant and underexplored challenge in medical vision-language modeling—generating radiology reports that are not only clinically accurate but also uncertainty-aware and reasoning-transparent. This focus is crucial for increasing trust and adoption of AI in clinical settings.

2. **Clear Writing**: The manuscript is clearly written and well-organized. The motivation, methodology, and experimental results are generally easy to follow, and the technical components are presented with sufficient clarity for readers to grasp the overall pipeline.

---

**Weaknesses:**

1. **Lack of Evaluation Against Frontier Models**: While the paper positions itself as addressing deficiencies in existing VLMs, it does not clearly establish whether state-of-the-art general-purpose models (e.g., GPT-4.5, Gemini, etc.) also suffer from the same issues. No direct evidence or comparative evaluation with such advanced models is provided, which weakens the empirical justification for the claimed gap.

2. **Insufficient Detail on Dataset Construction**: A critical part of the proposed framework is the creation of datasets annotated with uncertainty and reasoning information for supervised fine-tuning. However, the main text gives limited explanation of how these datasets (especially TRACE-CXR and the uncertainty-annotated set) were constructed. These are only briefly described in the appendix, which significantly reduces the reproducibility and credibility of this key contribution.

3. **Ambiguity in Reasoning Component**: The paper does not make it sufficiently clear whether the reasoning component is trained against reliable, human-authored reasoning ground truth or only LLM-generated pseudo-labels. This raises questions about the quality and validity of the reasoning supervision signal.

4. **Limited Baseline Comparisons**: Experimental comparisons are conducted primarily against a few relatively lightweight medical or vision-language models (e.g., MAIRA-2, HuatuoGPT-Vision-7B). The paper does not evaluate CURV against state-of-the-art generalist models that may already possess strong reasoning or uncertainty modeling capabilities. This limits the ability to assess the real advantage of CURV.

---

> ### Author Rebuttal · Authors · 2025-07-31
>
> We sincerely thank Reviewer for their constructive feedback. Based on the valuable comments from all reviewers, we have conducted **three significant new experiments and prepared detailed clarifications** to strengthen our work.
>
> **Summary of Major Updates**:
>
> - **New SOTA Comparison**: We performed a new evaluation against Gemini 2.5 Pro. The results confirm CURV's superior performance, with our model achieving a RadGraph F1 of 25.65 vs. Gemini's 7.71.
>
> - **New Clinical Expert Validation**: To validate our LLM-based evaluation, we conducted a new study with a board-certified radiologist. Their scores showed strong alignment with our LLM evaluators (e.g., MAE of 0.38 vs. Gemini 2.5 Pro), confirming our protocol's reliability.
>
> - **New Out-of-Distribution (OOD) Evaluation**: To test generalization, we ran a new evaluation on the IU X-ray dataset. CURV maintains its strong performance and outperforms all baselines, demonstrating that its learned capabilities are robust and not overfit to MIMIC-CXR.
>
> We address the reviewer's specific points below.
>
> # Weakness 1 & Weakness 4 & Question 1
>
> We sincerely thank the reviewer for this excellent suggestion. We agree that comparing CURV to frontier foundation models is a crucial test of its capabilities. As suggested, we conducted a new set of experiments evaluating a state-of-the-art generalist model, Gemini 2.5 Pro, on our test set. The results, summarized in the table below, confirm our central hypothesis: while powerful generalist models perform strongly on average, they lack the specialized mechanisms for generating clinically nuanced reports.
>
> *Table 1: Comparison between Gemini 2.5 pro and CURV on generation metrics*
> | **Model** | **B-1** | **B-2** | **B-3** | **B-4** | **METEOR** | **R-L** | **gritlm** |
> | :--- | :--- | :--- | :--- | :--- | :--- | :--- | :--- |
> | Gemini 2.5 pro | 12.54 | 5.20 | 2.25 | 1.05 | 21.19 | 15.01 | 40.41 |
> | CURV | **29.23** | **18.76** | **12.08** | **6.86** | **38.30** | **39.08** | **54.89** |
>
> *Table 2: Comparison between Gemini 2.5 pro and CURV on clinical accuracy metrics*
> | Model | f1chexbert | | | radgraph | | | |
> |:---|:---:|:---:|:---:|:---:|:---:|:---:|:---:|
> | | Acc. | Macro F1 | Micro F1 | Ent. F1 | F1 | Prec. | Rec. |
> | Gemini 2.5 pro | 74.35 | 5.35 | 48.45 | 13.09 | 7.71 | 26.76 | 11.56 |
> | CURV | **91.56** | **5.86** | **74.36** | **36.99** | **25.65** | **48.80** | **35.28** |
>
> As the results clearly show, CURV significantly outperforms Gemini 2.5 Pro across all metrics, especially in clinical accuracy (e.g., RadGraph F1 of 25.65 vs. 7.71). This new evidence demonstrates that generating high-quality, trustworthy radiology reports is a specialized task that benefits immensely from CURV's targeted framework.
>
> We will add this new baseline comparison with Gemini 2.5 Pro, along with this analysis, to the final version of our paper to further strengthen our claims and provide a more comprehensive evaluation.
>
> # Weakness 2 & Question 2
> We apologize that placing these details in the appendix reduced the paper's self-contained clarity. To address this, we will move the essential details of our dataset curation process from Appendix B into the main paper. This will improve the paper's clarity and reproducibility. For the uncertainty expression related details:
> 1.  **Alignment with Image Regions:** This process began by using the Imagenome dataset [1]. This dataset provides pre-existing links between specific textual phrases in MIMIC-CXR radiology reports and their corresponding bounding box localizations on chest X-ray images. This step established the alignment between a finding described in the text and a region on the image.
> 2.  **Extraction of Uncertainty Expressions:** With the text-to-image alignment established, we then used the Qwen2.5-7B-instruct model to perform an extraction of uncertainty language. The model was prompted to identify and pull out specific uncertainty-expressing phrases (e.g., "likely") directly from the report sentences that Imagenome had already linked to visual findings. This step maps linguistic uncertainty to precise anatomical regions. The prompt we used to extract uncertainty expression is: *"You are a medical text analyzer that extracts uncertainty expressions. Extract any words or phrases expressing uncertainty from the following medical text. Return ONLY the uncertainty expressions, one per line. If none exist, return NONE."*
>
> [1] J. Wu, et al. Chest imagenome dataset, 2021.
>
> # Weakness 3 & Questions 3 & 4
> We thank the reviewer for their insightful comment regarding the validation of our LLM-generated TRACE-CXR dataset. The reasoning traces are purely generated by LLMs. On one hand, the practice of generating data to achieve better model learning is becoming more common in recent research works. Yet, ensuring the quality of the generated data and how to use it properly is also vital. This valuable feedback prompted us to conduct a new, formal evaluation with a board-certified radiologist to rigorously assess our methodology. We elaborate our arguments and present our new experimental results as follows.
>
> **1. On the Validity of LLM-Generated Training Data**
> * **Precedent and Community Standard:** Using LLMs to generate training data is now standard practice, underpinning widely-used datasets like Alpaca [1] and WizardLM [2]. The seminal Self-Instruct [3] paper established that a model fine-tuned on purely synthetically generated instructions could match models trained on expensive, human-annotated data, demonstrating that the utility of a large synthetic dataset can overcome point-wise imperfections. Furthermore, recent studies like the Vision-R1 (2025) model [4] use "automated reasoning dataset construction" without human validation, confirming this is a state-of-the-art approach.
> * **RL as a Corrective Mechanism Grounded in Human Data:** Crucially, our SFT-then-RL pipeline is designed to mitigate risks from synthetic data. The LLM-generated TRACE-CXR dataset is only used during the SFT "reasoning initialization" stage (Stage 2). Its purpose is to teach the model the basic tripartite structure and provide a policy suitable for exploration. In the subsequent RL phase (Stage 3), the model does not make use of any ground-truth `<thinking>` text. Instead, it is rewarded for generating a coherent `<thinking>` section that connects well to the ground-truth, human-authored `<findings>` and `<impression>` sections from the MIMIC-CXR reports. This design means the model is not learning to simply mimic potentially flawed LLM text. Instead, the RL process forces the model to discover a good reasoning path, guided only by a coherence reward tied to genuine clinical endpoints. The RL stage thereby acts as an implicit, task-based validation and refinement mechanism, reducing the dependency on the quality of the initial SFT data.
>
> [1] Taori, et al. "Stanford alpaca: An instruction-following llama model." 30 Jun. 2023.
>
> [2] Xu, Can, et al. "WizardLM: Empowering large pre-trained language models to follow complex instructions." ICLR, 2024.
>
> [3] Wang, et al. "Self-Instruct: Aligning Language Models with Self-Generated Instructions." ACL, 2023.
>
> [4] Huang, et al. "Vision-r1: Incentivizing reasoning capability in multimodal large language models." *arXiv preprint arXiv:2503.06749* (2025).
>
> **2. New Clinical Evaluation: Comparing LLM Evaluators with an Expert Radiologist**
>
> *Table 3: Comprehensive comparison of evaluation scores between Clinicians (Clin), Grok3 (Grok), and Gemini 2.5 Pro (Gem).*
>
> | **Study ID** | **Logical Coherence** | | | **Depth of Analysis** | | | **Relevance** | | | **Evidence Based** | | | **Consistency** | | |
> |:---|:---:|:---:|:---:|:---:|:---:|:---:|:---:|:---:|:---:|:---:|:---:|:---:|:---:|:---:|:---:|
> | | **Clin** | **Grok** | **Gem** | **Clin** | **Grok** | **Gem** | **Clin** | **Grok** | **Gem** | **Clin** | **Grok** | **Gem** | **Clin** | **Grok** | **Gem** |
> | **s54517467** | 4 | 5 | 5 | 5 | 5 | 5 | 5 | 5 | 5 | 4 | 5 | 5 | 4 | 5 | 5 |
> | **s51966501** | 4 | 5 | 5 | 5 | 5 | 5 | 4 | 5 | 5 | 4 | 5 | 5 | 4 | 5 | 5 |
> | **s52188295** | 5 | 5 | 5 | 5 | 5 | 5 | 4 | 5 | 5 | 4 | 5 | 5 | 5 | 5 | 5 |
> | **s55493024** | 4 | 5 | 5 | 5 | 5 | 5 | 4 | 5 | 5 | 4 | 5 | 5 | 5 | 5 | 5 |
> | **s51074196** | 4 | 5 | 5 | 5 | 5 | 5 | 4 | 5 | 5 | 4 | 5 | 5 | 5 | 5 | 5 |
> | **s56689492** | 5 | 5 | 5 | 5 | 5 | 5 | 5 | 5 | 5 | 5 | 5 | 5 | 5 | 5 | 5 |
> | **s57002637** | 5 | 5 | 5 | 5 | 5 | 5 | 5 | 5 | 5 | 5 | 5 | 5 | 5 | 5 | 5 |
> | **s50849849** | 5 | 5 | 5 | 5 | 5 | 5 | 5 | 4 | 5 | 5 | 4 | 5 | 5 | 3 | 4 |
> | **s51215354** | 5 | 5 | 5 | 5 | 5 | 5 | 5 | 5 | 5 | 5 | 5 | 5 | 5 | 5 | 5 |
> | **s52528325** | 4 | 5 | 5 | 5 | 5 | 5 | 4 | 5 | 5 | 4 | 5 | 5 | 5 | 5 | 5 |
>
> We agree that direct empirical validation is the strongest form of evidence. To that end, we conducted a new formal evaluation with a board-certified radiologist to directly compare their judgments against our LLM-based evaluation protocol. To be maximally thorough, we tested two different SOTA LLMs (Grok3 and Gemini 2.5 Pro).
> * **Methodology**: A board-certified radiologist, Grok3, and Gemini 2.5 Pro scored the same set of 10 reports randomly selected in the TRACE-CXR dataset, using the exact same criteria from our paper (Logical Coherence, Consistency, etc.).
> * **Results**: As shown in Table 3, both LLMs show strong alignment with the human expert, confirming that our evaluation protocol is reliable. Both LLMs demonstrated a high degree of alignment. For gemini 2.5 pro, scores were within 1 point of the clinician's 100% of the time and with a very low Mean Absolute Error (MAE) of 0.380 compared to Clinicians. For Grok 3, scores were within 1 point of the clinician's 98.0% of the time, with a low MAE of 0.440.
>
> In summary, this new clinical evaluation, combined with our framework's inherent methodological safeguards, provides strong, multi-faceted validation for our approach and confirms that our LLM-based metrics are a reliable proxy for expert judgment.

---

> > ### Comment · Reviewer_e8n6 · 2025-08-06
> >
> > I appreciate the authors' rebuttal, which has addressed most of my concerns. At this stage, I intend to maintain my original score and would be interested in seeing the feedback from the other reviewers.

---

### Official Review · Reviewer_ZQB5 · 2025-07-03

**Clarity:** 2
**Significance:** 2
**Originality:** 2
**Rating:** 4
**Confidence:** 3

**Summary:**

This paper proposes CURV, a novel vision-language model (VLM) framework for radiology report generation from chest X-ray (CXR) images. CURV targets two critical limitations of existing VLMs in the clinical setting: (1) a lack of uncertainty awareness, and (2) the absence of explicit reasoning that connects image findings to diagnostic impressions. The CURV framework introduces three main components: First, uncertainty modeling – captures both structural (finding-level) and semantic (impression-level) diagnostic uncertainty; second, structured reasoning generation – includes an intermediate $<thinking>$ section to explain the reasoning from findings to impressions; thrid, reinforcement learning (RL) – uses Group Relative Policy Optimization (GRPO) with a multi-dimensional reward function to improve coherence and clinical quality. Experimental results on the MIMIC-CXR dataset demonstrate that CURV outperforms several baselines (e.g., LLaVA-1.5-7B, MAIRA-2) in both standard text generation metrics (e.g., BLEU, ROUGE) and clinical accuracy scores (e.g., CheXbert, RadGraph F1).

**Questions:**

Questions
1.	How does RL refine the reasoning capability beyond supervised learning? Could the author provide more clarification on how the dataset is used in the supervised finetuning and RL finetuning? If the dataset already provided the thinking chain, would this make the RL a fully supervised training?
2.	Could the author provide more details around the training hyper-parameters and training settings?
3.	Any plans for expert evaluation? LLM-based evaluations are helpful, but would the authors consider radiologist review in future iterations?

**Ethical Concerns:**

["NO or VERY MINOR ethics concerns only"]

**Final Justification:**

During the rebuttal period, the authors resolved all of my concerns regarding SFT, RL, and computation resources. Therefore, I will increase my scores to borderline accept.

**Limitations:**

Yes

**Quality:**

2

**Strengths And Weaknesses:**

Strengths
- Significant clinical motivation: The paper clearly identifies two critical missing components in current VLMs—structured uncertainty and transparent reasoning—which are essential for real-world trust and adoption in medicine.
- New dataset contribution: The authors introduce TRACE-CXR, a curated dataset of 2,000 CXR reports augmented with explicit LLM-generated reasoning paths, which is valuable for future work on clinical explainability.
- Strong empirical results: CURV achieves impressive improvements over both base and fine-tuned large models, despite using a relatively small 3B backbone.

Weaknesses
- Clarity around the RL phase is lacking: If the reasoning path is already given during supervised pretraining, what exactly is being learned or discovered during RL? The TRACE-CXR contains full thinking process. Would this make the training process simply a fully supervised finetuning? How exactly the dataset and which dataset is for the RL finetuning?
- Possible contradiction in supervision vs. self-discovery: The design implies that the model learns to “reason” from examples rather than discovering reasoning paths on its own, which slightly undermines the RL-based reasoning enhancement claim.
- The training hyper-parameters and training settings needs more clarification: The model is finetuned using four A100 GPUs, which in my experience is difficult to fit in a 3B model. I am wondering if LoRA is more realistic than the full model finetuning.

---

> ### Author Rebuttal · Authors · 2025-07-31
>
> We thank Reviewer ZQB5 for their thoughtful review and for recognizing our work's "significant clinical motivation" and "strong empirical results". Based on the valuable comments from all reviewers, we have conducted **three significant new experiments and prepared detailed clarifications** to strengthen our work.
>
> **Summary of Major Updates**:
>
> - **New SOTA Comparison**: We performed a new evaluation against Gemini 2.5 Pro. The results confirm CURV's superior performance, with our model achieving a RadGraph F1 of 25.65 vs. Gemini's 7.71.
>
> - **New Clinical Expert Validation**: To validate our LLM-based evaluation, we conducted a new study with a board-certified radiologist. Their scores showed strong alignment with our LLM evaluators (e.g., MAE of 0.38 vs. Gemini 2.5 Pro), confirming our protocol's reliability.
>
> - **New Out-of-Distribution (OOD) Evaluation**: To test generalization, we ran a new evaluation on the IU X-ray dataset. CURV maintains its strong performance and outperforms all baselines, demonstrating that its learned capabilities are robust and not overfit to MIMIC-CXR.
>
> We address the reviewer's specific points below.
>
> # Weakness 1 & Weakness 2 & Question 1
> We sincerely apologize for the lack of clarity on this point. We have two different datasets for the SFT and RL stages, where the reasoning path is only used for the initial SFT. The details of the SFT and RL processes are further clarified as follows:
>
> * **Stage 2 (SFT Reasoning Initialization):** In this stage, the model is fine-tuned based on the complete TRACE-CXR dataset, which includes `<findings>`, `<thinking>`, and `<impression>` sections. This supervised learning step aims to teach the model the basic structure, providing it with a foundational template for what a "reasoning" section looks like.
> * **Stage 3 (RL Enhancement):** For the RL phase, the model is provided with only the `<findings>` and `<impression>` sections from the MIMIC-CXR dataset, while the ground-truth `<thinking>` text is not used. The model explores how to discover a better reasoning path guided by the reward signals.
>
> Our SFT-then-RL methodology aligns well with the recent studies indicating that relying solely on SFT can cause models to memorize reasoning patterns and fail to generalize [1]. Our RL stage avoids this imitation trap. More importantly, by withholding the ground-truth `<thinking>` section, our approach prevents the "pseudo reasoning" problem where models rigidly imitate suboptimal paths from the SFT data [2]. This process forces the model to move beyond simple mimicry and learn to generate functionally coherent reasoning.
>
> [1] Chu, Tianzhe, et al. "SFT Memorizes, RL Generalizes: A Comparative Study of Foundation Model Post-training." *Forty-second International Conference on Machine Learning.*
>
> [2] Chen, Hardy, et al. "Sft or rl? an early investigation into training r1-like reasoning large vision-language models." *arXiv preprint arXiv:2504.11468* (2025).
>
> ---
>
> # Weakness 3 & Question 2
>
> We thank the reviewer for this suggestion to improve reproducibility. While full-parameter tuning of a 3B model is indeed memory-intensive, it is entirely feasible on a 4x A100-80G node by leveraging a combination of standard, state-of-the-art memory optimization techniques.
>
> Our training approach was as follows:
>
> For Stages 1 & 2 (SFT), to maximize efficiency while adapting the model to the new data formats, we kept the vision encoder frozen and fine-tuned only the multimodal projection layer and the LLM backbone.
>
> For Stage 3 (RL), where the goal was to refine the entire model's behavior based on global reward signals, we performed full-parameter fine-tuning of the model checkpoint from Stage 2. To make this feasible, we employed several key techniques:
> * **Mixed-Precision Training**: We used bfloat16 precision, which halves the memory requirement for model parameters and gradients compared to standard FP32, with minimal impact on training stability.
> * **Optimizer State and Parameter Sharding**: The primary memory bottleneck is often not the model weights, but the optimizer states (e.g., Adam's momentum and variance buffers, which can be 6x the size of the parameters in FP32) and activations. A naive calculation for a 3B model in mixed precision requires approximately 48 GB of VRAM (6 GB for fp16 parameters + 6 GB for gradients + 36 GB for fp32 optimizer states and weight copies). We utilized DeepSpeed with ZeRO Stage 3 optimization. This powerful technique partitions not only the optimizer states but also the gradients and the model parameters themselves across all 4 GPUs. This means each GPU only holds a fraction (~1/4) of the total training-related tensors. This reduces the per-GPU memory requirement from ~48 GB to a much more manageable ~12 GB, which fits comfortably within the 80 GB VRAM of each A100 GPU.
> * **Activation Checkpointing**: The second major memory bottleneck comes from storing activations (the intermediate outputs of each layer), the size of which scales with batch_size * context_length * hidden_size * num_layers. To manage this context-dependent memory usage, we employed activation checkpointing (also known as gradient checkpointing). Instead of storing all activations in memory during the forward pass, this technique discards them and recomputes them during the backward pass where needed. This strategy trades a modest amount of additional compute time for a very significant reduction in activation memory, ensuring that even with a long context length, the memory footprint remains manageable.
>
> The combination of bfloat16, DeepSpeed ZeRO-3, and activation checkpointing is a standard and robust strategy for full-parameter fine-tuning of multi-billion parameter models, and it allows a 3B model to train comfortably within the 80GB of memory available on each A100 GPU. We will add these specific details about our use of bfloat16 precision, DeepSpeed ZeRO-3, and activation checkpointing to Appendix A in the final version to ensure full reproducibility.
>
> ---
>
> # Question 3
>
> This is an excellent point. We completely agree that a formal evaluation by radiologists is the gold standard and a critical next step for this research. While our automated and LLM-based metrics serve as valuable proxies for development, validating the true clinical utility and safety of CURV-generated reports requires human expert review. This was beyond the scope of this work but is a top priority for our future research agenda. We will add the Limitations section of our paper to explicitly state this and discuss our plans to conduct such an evaluation.

---

> > ### Comment · Reviewer_ZQB5 · 2025-08-06
> >
> > I appreciate the detailed response from the authors. The response clarified SFT, RL, and computation resources related questions. And I don't have further comments.

---

### Note · Authors · 2025-08-15

We sincerely thank the reviewers and the Area Chair for their time and insightful feedback, which has been invaluable in strengthening our paper.

The reviewers' primary concerns centered on three key areas: evaluation, data validity, and technical clarity. We addressed these head-on with **three significant new experiments** and detailed clarifications:

1. **New SOTA Comparison**: As requested (R-e8n6), we evaluated against Gemini 2.5 Pro. The results confirmed CURV's superior clinical accuracy (e.g., RadGraph F1 of 25.65 vs. 7.71), demonstrating the value of our specialized approach.

2. **New Clinical Expert Validation**: To validate our LLM-based methodology (a concern of R-e8n6, R-Nede), we performed a new study with a board-certified radiologist. Their evaluations showed strong alignment with our protocol (MAE of 0.38), confirming its reliability.

3. **New Out-of-Distribution (OOD) Evaluation**: To test robustness (a key point from R-qpYn), we evaluated CURV on the IU X-ray dataset. Our method maintained its strong performance, demonstrating that its capabilities generalize well.

We are encouraged that reviewers recognized our work's "significant clinical motivation" (R-ZQB5) and are pleased that our new evidence has "fully resolved" the concerns of R-qpYn and satisfied R-ZQB5. We hope our new radiologist validation, combined with our clarification on the SFT-then-RL framework (where RL grounds the model in human-authored clinical data), decisively addresses the central questions about our methodology.

We have a clear plan to integrate these new results and all other promised revisions into the final manuscript. We are confident the revised paper will be a strong contribution to the NeurIPS community by advancing trustworthy, explainable AI in a high-stakes clinical domain. Thank you for your consideration.

---

### Decision · Program_Chairs · 2025-09-17

**Decision:**

Accept (poster)

**Comment:**

The paper gets reviewing comments from four experts. Three experts are positive or slightly positive about the paper and one expert (Reviewer e8n6) is slightly negative.

Reviewer e8n6 also admits that his original concerns are mostly answered by the rebuttal and is willing to hear the feedbacks from other reviewers, who are all positve.  Thus, the AC decides to downweigh his/her comments and recommends to accept the paper as a poster.